# Competition between histone and transcription factor binding regulates the onset of transcription in zebrafish embryos

Shai R Joseph[1], Máté Pálfy[1], Lennart Hilbert[1,2,3], Mukesh Kumar[1], Jens Karschau[3], Vasily Zaburdaev[2,3], Andrej Shevchenko[1], Nadine L Vastenhouw[1]*

[1]Max Planck Institute of Molecular Cell Biology and Genetics, Dresden, Germany; [2]Center for Systems Biology Dresden, Dresden, Germany; [3]Max Planck Institute for the Physics of Complex Systems, Dresden, Germany

**Abstract** Upon fertilization, the genome of animal embryos remains transcriptionally inactive until the maternal-to-zygotic transition. At this time, the embryo takes control of its development and transcription begins. How the onset of zygotic transcription is regulated remains unclear. Here, we show that a dynamic competition for DNA binding between nucleosome-forming histones and transcription factors regulates zebrafish genome activation. Taking a quantitative approach, we found that the concentration of non-DNA-bound core histones sets the time for the onset of transcription. The reduction in nuclear histone concentration that coincides with genome activation does not affect nucleosome density on DNA, but allows transcription factors to compete successfully for DNA binding. In agreement with this, transcription factor binding is sensitive to histone levels and the concentration of transcription factors also affects the time of transcription. Our results demonstrate that the relative levels of histones and transcription factors regulate the onset of transcription in the embryo.

**\*For correspondence:**
vastenhouw@mpi-cbg.de

**Competing interests:** The authors declare that no competing interests exist.

## Introduction

In many organisms, early embryonic development is directed exclusively by maternal products that are deposited into the female gamete during oogenesis. Following the clearance of a subset of these products (*Yartseva and Giraldez, 2015*), transcription is initiated and the zygotic genome acquires developmental control (*Blythe and Wieschaus, 2015a*; *Harrison and Eisen, 2015*; *Lee et al., 2014*; *Tadros and Lipshitz, 2009*). This handover is referred to as the maternal-to-zygotic transition and the onset of transcription is called zygotic genome activation (ZGA). The absolute time and number of cell cycles required before the first transcripts can be detected is species specific (*Tadros and Lipshitz, 2009*). Additionally, from one gene to another the timing of transcriptional activation varies (*Aanes et al., 2011*; *Collart et al., 2014*; *Harvey et al., 2013*; *Heyn et al., 2014*; *Lott et al., 2011*; *Owens et al., 2016*; *Pauli et al., 2012*; *Sandler and Stathopoulos, 2016*; *Tan et al., 2013*). In fact, for some genes the first zygotic transcripts can be detected several cell cycles before the stage that is traditionally defined as the time point of ZGA (*De Renzis et al., 2007*; *Heyn et al., 2014*; *Skirkanich et al., 2011*; *Yang et al., 2002*). In spite of the progress made, it remains unclear how the onset of transcription in embryos is temporally regulated.

Several lines of evidence suggest that the absence of transcription during early embryonic development could be due to limited levels of transcription factors (*Almouzni and Wolffe, 1995*; *Veenstra et al., 1999*). In this scenario, transcriptional activation would occur once a threshold level

**eLife digest** The DNA in a fertilized egg contains all the information required to form an animal's body. In order for the animal to develop properly, particular genes encoded in the DNA are only active at specific times. The DNA is wrapped around proteins called histones, which allows the DNA to be tightly packed inside the cell. However, histones can block other proteins called transcription factors from binding to the DNA to activate the genes. Young embryos initially develop with all of their genes switched off, relying on the nutrients and other molecules provided by their mother. After some time, the embryo starts to switch on its own genes to take control of its own development, but it was not clear how this happens.

Joseph et al. investigated how genes are activated in zebrafish embryos, which are often used as models to study how animals develop. The experiments show that competition between histones and transcription factors for binding to DNA controls when genes are switched on. In young fish embryos, there are so many histones present that transcription factors have no opportunity to bind to DNA. Over time, however, the numbers of histones decrease, allowing transcription factors to bind to DNA and switch on genes.

Histones and transcription factors regulate the activity of genes throughout the life of the animal. Therefore, competition between these two types of protein may also control gene activity in other situations. A better understanding of how gene activity is controlled could allow researchers to more easily grow different types of cell in the laboratory or to reprogram specific cells in the body. As such, these new findings may aid the development of therapies to regenerate organs or tissues that have been damaged by injury or disease.

of these factors is reached. For example, experiments that used the transcriptional activity of injected plasmids as a read-out revealed that an increase in the amount of the potent, heterologous, transcriptional activator GAL4-VP16 can overcome transcriptional repression of its target gene in the early embryo (*Almouzni and Wolffe, 1995*). However, it remained unclear whether limited levels of transcription factors contribute to the absence of endogenous transcription in early embryos. Additional support for the limited machinery model came from work showing that an increase in the concentration of the general transcription factor TBP can cause premature transcription from an injected – and incompletely chromatinized – DNA template in *Xenopus* embryos. This effect was maintained only when non-specific DNA was added to titrate chromatin assembly (*Almouzni and Wolffe, 1995*; *Veenstra et al., 1999*). These results suggested that low TBP levels may play a role in the absence of transcription during the early stages of *Xenopus* development, but that increasing TBP alone is not sufficient to cause sustained premature transcription. During the cleavage stages of *Xenopus* development, TBP levels increase due to translation, which suggests that TBP levels might contribute to the timely activation of transcription during ZGA (*Veenstra et al., 1999*). Transcription factors have recently been identified that are required for the activation of the first zygotically expressed genes in *Drosophila* (Zelda) and zebrafish (Pou5f3, Sox19b, Nanog) (*Harrison et al., 2011*; *Lee et al., 2013*; *Leichsenring et al., 2013*; *Liang et al., 2008*; *Nien et al., 2011*). RNA for these factors is maternally provided and their levels increase due to translation during the early cell cycles. This suggests the possibility that an increase in the concentration of these transcription factors might contribute to the shift from transcriptional repression to transcriptional activity. Although transcription factors levels clearly influence transcriptional activity during early embryogenesis, there is evidence to show that the transcriptional machinery is operational prior to ZGA (*Dekens et al., 2003*; *Lu et al., 2009*; *Newport and Kirschner, 1982a*, *1982b*; *Prioleau et al., 1994*) (see below). Thus, the timing of ZGA cannot be solely explained by a requirement to reach a threshold level of transcriptional activators.

The finding that a premature increase in the number of nuclei or the amount of DNA resulted in premature transcription of injected plasmids in *Xenopus* embryos suggested that the transcriptional machinery is fully functional prior to genome activation and led to the excess repressor model (*Newport and Kirschner, 1982a*). This model postulates that a transcriptional repressor is titrated by binding to the exponentially increasing amount of genomic DNA, until it is depleted first from

the soluble fraction, and then from DNA, to allow for the onset of transcription. Related studies in zebrafish and *Drosophila* have provided further evidence for this model. Endogenous transcription is initiated earlier in zebrafish embryos that accumulate DNA due to a defect in chromosome segregation (*Dekens et al., 2003*), and transcription is delayed in haploid *Drosophila* embryos compared to diploid embryos, albeit not for all genes (*Lu et al., 2009*). The excess repressor model predicts that the repressor is present in large excess, at relatively stable levels while the genome is inactive, and can bind DNA with high affinity. Core histones fulfill these criteria (*Adamson and Woodland, 1974*; *Woodland and Adamson, 1977*). Moreover, when bound to DNA in the form of nucleosomes, histones can affect DNA accessibility for DNA-binding proteins. To date, two key studies have investigated the role of core histones in the temporal regulation of zygotic transcription in *Xenopus* embryos (*Almouzni and Wolffe, 1995*; *Amodeo et al., 2015*). Experiments that used the transcriptional activity of injected plasmids as a read-out revealed that premature transcription caused by an excess of non-specific DNA can be negated by the addition of histones (*Almouzni and Wolffe, 1995*). More recently, the level of histones H3/H4 was shown to regulate the level of transcription in *Xenopus* egg extract and H3 was suggested to play a similar role in the embryo (*Amodeo et al., 2015*). Taken together, these results support the idea that histones play a role in regulating the timing of zygotic transcription.

If histones function as repressors according to the original excess repressor model, it would be predicted that a substantial reduction of the histone-density on DNA would cause the onset of transcription (*Amodeo et al., 2015*; *Newport and Kirschner, 1982a*). However, while such a scenario might be possible for typical sequence-specific repressors of transcription, it is unlikely for histones. Histones assemble into histone octamers on DNA to form nucleosomes, the basic building blocks of chromatin. Thus, random depletion of nucleosomes from DNA would severely compromise the integrity of chromatin structure. Taken together, there is support for the idea that histone levels play a role in regulating the timing of zygotic transcription, but it remains unclear how this would mechanistically work. Furthermore, the observation that both activator and histone levels play a role in shifting the balance between repression and activation at genome activation remains to be clarified.

Here, we analyze the onset of zygotic transcription in zebrafish embryos. With a quantitative approach, we show that the concentration of non-DNA-bound histones determines the timing of zygotic transcription and that all four core histones are required for this effect. The reduction in nuclear histone concentration that coincides with genome activation does not result in a significant change in nucleosome density, but rather allows transcription factors to successfully compete for DNA binding. In agreement with this, the association of transcription factors with the genome is sensitive to histone levels, and changing the concentration of transcription factors also affects the time of transcription. Our results show that transcription is regulated by a dynamic competition for DNA binding between histones and transcription factors. Transcription begins when the concentration of non-DNA-bound histones in the nucleus has sufficiently dropped so that the transcriptional machinery can outcompete histones for binding to DNA.

## Results

In zebrafish, zygotic transcription starts ~3 hr post-fertilization, around the tenth cell division (*Aanes et al., 2011*; *Harvey et al., 2013*; *Heyn et al., 2014*; *Kane and Kimmel, 1993*; *Pauli et al., 2012*) (*Figure 1A*). To analyze the onset of transcription in the embryo in detail, we identified six genes that are not maternally provided and that have previously been shown to be activated at the onset of genome activation (*Aanes et al., 2011*; *Pauli et al., 2012*) (*Figure 1—figure supplement 1A*). At the 1000-cell (1K) stage, transcripts can be detected, especially when choosing late stage embryos (*Figure 1B*). Thus, to clearly distinguish between transcription being off and on, we analyzed early 1K and (mid) high stage embryos. Using this approach, analysis by qPCR allowed us to detect consistent induction of these genes at high stage (*Figure 1B* and *Figure 1—figure supplement 1B*). We will refer to the stages before and after induction as before and following genome activation. To relate the onset of transcription to the number of cells present in the embryo, we next counted the number of cells in embryos ranging from 1K to dome stage. We imaged DAPI-stained nuclei on a two-photon microscope and counted them using the software Imaris (*Figure 1C*). Using nuclei as a proxy for cell number, we obtained counts that agreed with numbers of cells per embryo obtained by others for 1K (*Keller et al., 2008*). In contrast, our cell count for high stage, for

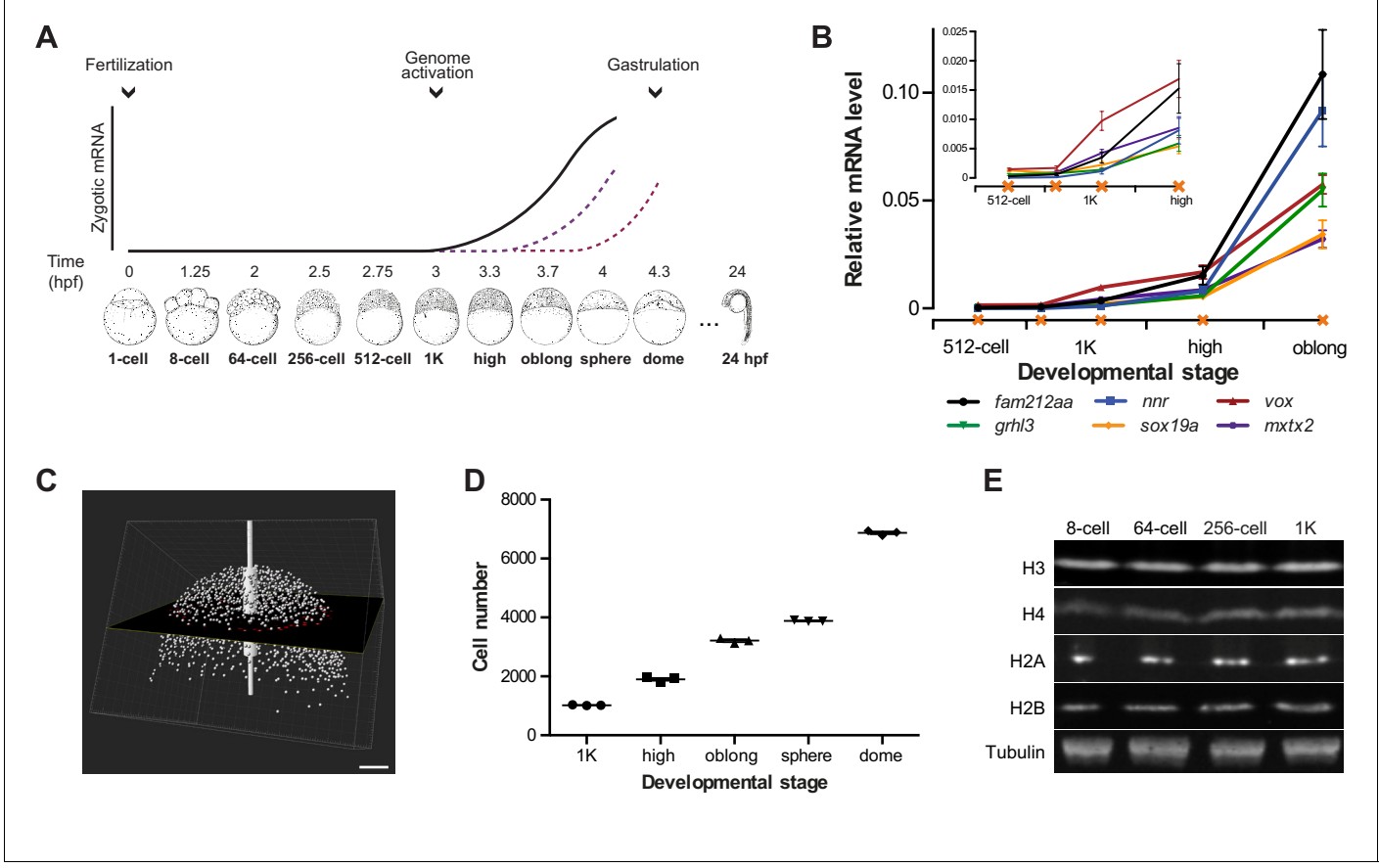

**Figure 1.** Assay to analyze the onset of transcription in zebrafish. (A) In zebrafish, transcription begins ~3 hr post-fertilization. Stage-specific drawings of representative embryos are adapted from *Kimmel et al. (1995)* with permission. (B) Expression of six genes was analyzed by qPCR at 512-cell, early 1K, mid 1K, high and oblong stage in wild-type embryos. Inset shows the same data, focusing on 1K and high stage. Data were taken from *Figures 2B* and *3D*. Error bars represent SEM (n ≥ 4). (C) Cell counting of DAPI-stained nuclei imaged with a two-photon microscope. Multiple z-slices and four tiles were stitched to allow rendering in imaging software Imaris and the calculation of cell number. Image shown is a representative example of high stage. Scale bar, 100 μm. (D) Quantification of the number of cells at 1K, high, oblong, sphere and dome stage. Each data point represents a biological replicate consisting of three embryos. Error bars represent SEM (n = 3). (E) Western blots showing the protein level of histone H3, H4, H2A, and H2B in embryos at 8-cell, 64-cell, 256-cell and 1K stages. Tubulin was used to control for equal loading in each experiment. Blots shown are representative examples (n > 3).

*Figure 1* part A, lower panel sketches of embryos were reproduced from *Kimmel et al. (1995)* with permission (© copyright John Wiley and Sons, 1995. All Rights Reserved).

The following source data and figure supplement are available for figure 1:

**Source data 1.** Cell numbers for wild-type zebrafish embryos.

**Figure supplement 1.** Selection of genes and *eif4g2α* as a normalizer gene.

example, was slightly higher (1900 vs 1800) (*Keller et al., 2008*), which is consistent with the later high stage which we analyzed (*Figure 1D* and *Figure 1—source data 1*). We conclude that the set of genes we selected is robustly induced at high stage, when embryos contain ~1900 cells, and thus represent a reliable system to analyze the onset of zygotic transcription during embryogenesis.

## Increasing the levels of all core histones delays onset transcription and gastrulation

Experiments in *Xenopus* embryos led to the hypothesis that histone levels regulate the onset of zygotic transcription (*Almouzni and Wolffe, 1995*; *Amodeo et al., 2015*). To analyze whether in

zebrafish, histones are potential candidates to be excess repressors, we analyzed the relative levels of the core histones—H3, H4, H2A and H2B—by Western blot. We found that they are present at relatively stable amounts from 8-cell to 1K stage (*Figure 1E*). Assuming that at 1K stage there are sufficient histones to wrap all genomes into chromatin, this suggests that histones are in excess relative to the amount of DNA during the earlier stages. Thus, histones could function as excess repressors of the zygotic genome in zebrafish.

If histones function as excess repressors in zebrafish embryos, it would be predicted that their level would affect the onset of transcription. To test this, we analyzed the effect of increasing the amount of histones in the embryo on the timing of transcriptional activation. We injected a stoichiometric mixture of the four core histones (from here on referred to as histone cocktail, HC; see Materials and methods for more details) into embryos at the 1-cell stage and then analyzed the onset of transcription for the previously characterized set of genes (*Figures 2A* and *1B*). An increase in the amount of histones delayed the onset of transcriptional activation: transcripts were detected at high stage in uninjected embryos, whereas in embryos injected with histone cocktail, transcription was only induced at oblong stage, a complete developmental stage later (*Figure 2B* and *Figure 2—figure supplement 1A*). Comparison of gene expression levels in uninjected and injected embryos at high stage (when transcripts can consistently be detected in uninjected embryos) revealed that the level of induction is reduced significantly upon injection of the histone cocktail but not upon injection of BSA as a control (*Figure 2B* and *Figure 2—figure supplement 1A*, bar graphs). Extending the analysis further, a large set of genes in Nanostring analysis confirmed that the effect we observed is general, and not limited to six genes (*Figure 2C*, *Figure 2—figure supplement 2*, *Figure 2—source data 1*). Staging by morphology was corroborated by cell counting, with absolute time between the analyzed stages being constant, confirming that changes in the timing of transcription were not due to effects on cell cycle length or developmental progression (*Figure 2—figure supplement 1B*). Moreover, the injected histones can be incorporated into chromatin, as indicated by labeling one of them with Cy5 and detecting this label in chromatin when imaging embryos after injection (*Figure 2—figure supplement 1C*), confirming that they are functional. Together, these data show that an increase in the excess amount of histones in the embryo delays the onset of transcription.

To test whether the effect we observe upon injecting the histone cocktail required an increase in the level of all four core histones, we next removed one histone at a time from the cocktail. The total protein content was kept constant by raising the level of the other three histones. Removing any histone from the histone cocktail impaired the ability of the histone cocktail to delay the onset of transcription (*Figure 2D* and *Figure 2—figure supplement 1D*). These results show that the injection of basic proteins into the embryo per se does not affect the onset of transcription. Moreover, these results argue that the effect of the histone cocktail relies on increasing the amount of all four histones and suggest that histones exert their repressive effect together.

Since the onset of zygotic transcription is known to be required for gastrulation (*Kane et al., 1996*; *Lee et al., 2013*; *Zamir et al., 1997*), we analyzed the effect of injecting the histone cocktail on the onset of gastrulation. Embryos injected with the histone cocktail initiated gastrulation later than uninjected embryos (*Figure 2E*). Although there was a delay following injection of BSA, it was significantly shorter than that observed with the histone cocktail and appeared to be a non-specific effect of injection (*Figure 2F*). Following the onset of gastrulation, embryos appeared to develop normally (*Figure 2E*, 24 hpf). Removing any histone from the histone cocktail reduced the developmental delay we observed upon injecting the histone cocktail (*Figure 2F*). We note that the developmental delay in the minus-one histone experiments was not reduced to the level observed for BSA injections. We therefore expect that injecting histones has an additional effect on developmental progression that is independent of the delay in transcription. We conclude that the delay in transcription as a consequence of increased histone levels causes a delay in the onset of gastrulation.

## Decreasing the pool of available histones causes premature transcription

If the level of histones regulates zygotic genome activation, it can also be predicted that a reduction in histone levels would result in the premature induction of transcription. A large fraction of the histones that is present at the onset of transcription is loaded in the egg already as protein (*Figure 1E*). Pentraxin three is a soluble pattern recognition molecule that has been shown to rapidly and irreversibly bind to the core histones H3 and H4 (*Bottazzi et al., 2010*; *Daigo et al., 2014*). We

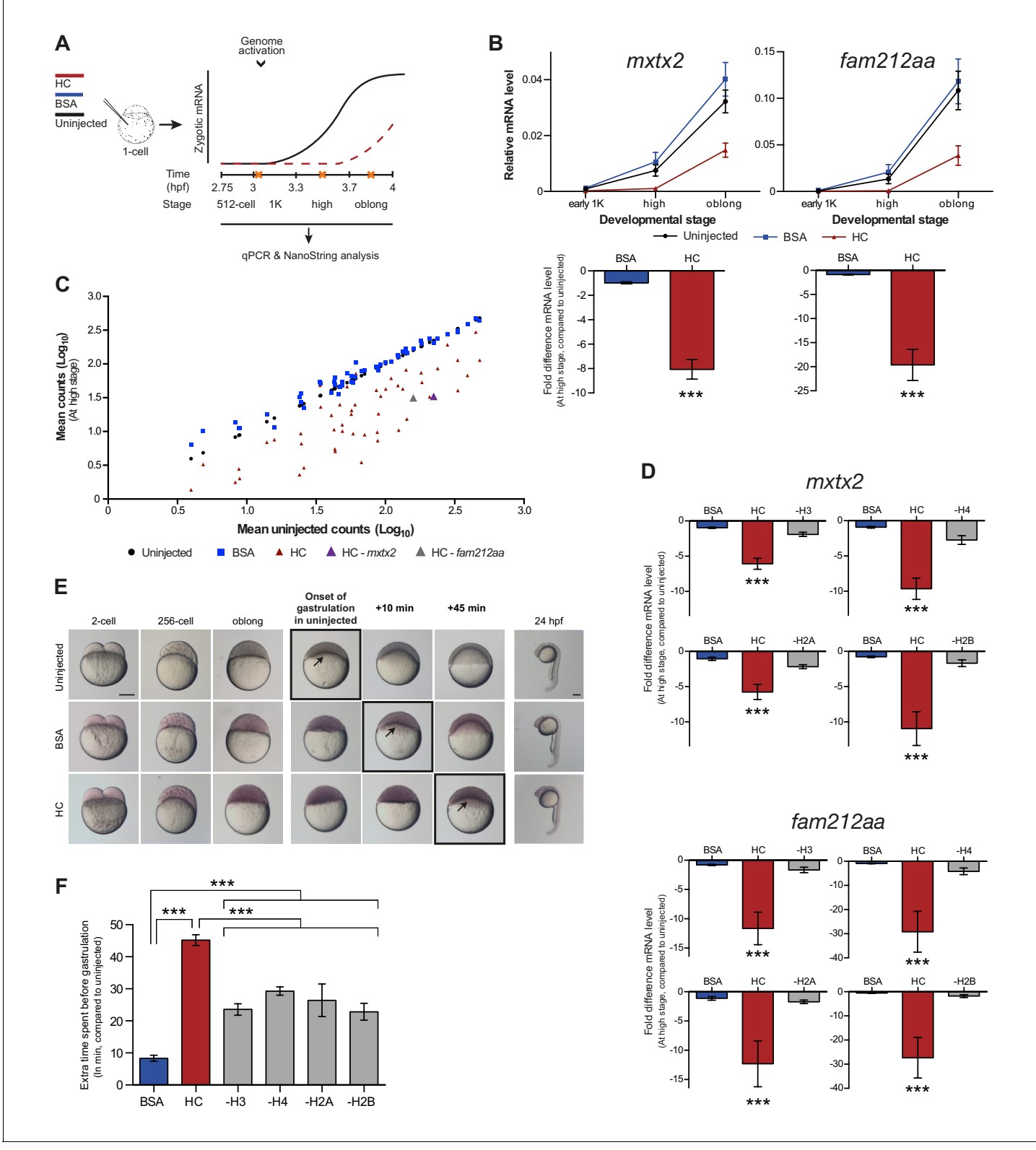

**Figure 2.** Increasing the levels of all core histones delays onset of transcription and gastrulation. (A) Schematic representation of experimental procedure. Histone cocktail (HC) containing ~5800 genomes worth of histones, or BSA was injected into the yolk of 1-cell embryos and qPCR and NanoString analysis was carried out at stages around genome activation. Orange crosses represent the timing of stages used for the analysis. (B) Expression of *mxtx2* and *fam212aa* was analyzed by qPCR at early 1K, high, and oblong stage in uninjected, BSA-injected and HC-injected embryos.
*Figure 2 continued on next page*

*Figure 2 continued*

Bar graphs show the same data, focusing on high stage. Error bars represent SEM (n ≥ 13). ***p<0.001 (two-tailed Student's t test, compared to BSA control). (C) Expression of 53 zygotically expressed genes was analyzed by NanoString analysis at high stage in uninjected, BSA-injected and HC-injected embryos. Mean counts of three independent biological replicates are shown. Location of *mxtx2* and *fam212aa* counts is indicated (See *Figure 2—figure supplement 2* for more details). (D) Relative expression level of *mxtx2* and *fam212aa* at high stage, for embryos injected with BSA, HC, and HC minus H3, H4, H2A, or H2B. Error bars represent SEM (n = 7). ***p<0.001 (ordinary one-way ANOVA). (E) Brightfield images of embryos that were not injected, injected with BSA, or injected with HC. Boxed images represent the onset of gastrulation. Scale bar shown for the uninjected 2-cell embryo applies to all treatments except for 24 hpf embryos which have a different scale bar. All scale bars represent 250 µm. hpf, hours post-fertilization. (F) Bar graph shows the quantification of the extra time it takes embryos to start gastrulation upon injecting BSA, HC, or HC minus one histone, compared to uninjected embryos. Error bars represent SEM (n = 27 for BSA, n = 25 for HC, n = 7 for HC minus one histone experiments). ***p<0.001 (ordinary one-way ANOVA with Tukey's multiple comparison test). In B and D, mRNA levels are normalized to the expression of *eif4g2α*.

The following source data and figure supplements are available for figure 2:

**Source data 1.** NanoString probe set.

**Figure supplement 1.** Increasing the levels of all core histones delays onset of transcription.

**Figure supplement 2.** Increasing the levels of all core histones delays onset of transcription for a large number of genes.

injected mRNA encoding PTX3 fused to RFP, to reduce the pool of available histones H3 and H4 in the zebrafish embryo (*Figure 3A*). As expected, total levels of H3 and H4 were not affected upon injection of this fusion construct (*Figure 3B*). Next, we examined if H4 co-precipitated with RFP-tagged PTX3 (*Figure 3C*). Indeed, this histone associates with PTX3 in vivo, suggesting that the injection of PTX3 results in a reduction of the soluble amount of histones H3 and H4 in zebrafish cells. A decrease in the soluble amount of histones caused premature transcription activation: transcripts were detected at early 1K stage, while in the uninjected embryos, transcripts were only detected at mid 1K (*Figure 3D* and *Figure 3—figure supplement 1A*). We included embryos at mid 1K in this experiment, in order to increase our resolution for detecting changes in transcription. Comparison of gene expression levels in uninjected and *ptx3*-injected embryos at early 1K stage (one time-point prior to when genes are first induced in uninjected embryos) revealed that the level of expression is increased upon injection of *ptx3* mRNA (*Figure 3D* and *Figure 3—figure supplement 1A*, bar graphs). A control injection with *rfp* mRNA did not result in co-precipitation with H4 (*Figure 3C*), nor did it affect the onset of transcription (*Figure 3D* and *Figure 3—figure supplement 1A*). Staging by morphology was corroborated by cell counting (*Figure 3—figure supplement 1B*), with absolute time between the analyzed stages being constant. Taken together, our results provide evidence that the level of core histones in the embryo dictates the timing of transcriptional activation.

## Onset of transcription coincides with a reduction in nuclear histone concentration

If histones were to function as excess repressors according to the original excess repressor model, the concentration of non-DNA-bound histones would be predicted to decrease during the cleavage stages of development. To test this prediction, we determined the absolute (molar) content and, correspondingly, the number of molecules of core histones in embryos using a quantitative mass spectrometry approach we recently developed (Kumar et al., unpublished) (*Figure 4A* and Materials and methods for more details). We analyzed embryos ranging from 1-cell to shield stage, when gastrulation is well underway (*Figure 4B* and *Figure 4—source data 1*). We observed an increase in the levels of histone protein until 1K stage, after which levels remained reasonably stable until sphere stage. Then, a rapid increase was observed, which is most likely the result of translation of zygotically produced histone mRNAs. Knowing the absolute numbers of histones per embryo as well as the calculated number of histones required to wrap a genome (*Figure 4B*, *Figre 4—source data 1*), allowed us to derive the number of genomes worth of histones per embryo. From that, we derived the number of excess (non-DNA bound) histones per cell in genomes worth of histones, for embryos ranging from 1-cell to dome stage (*Figure 4C* and Materials and methods for more details). For

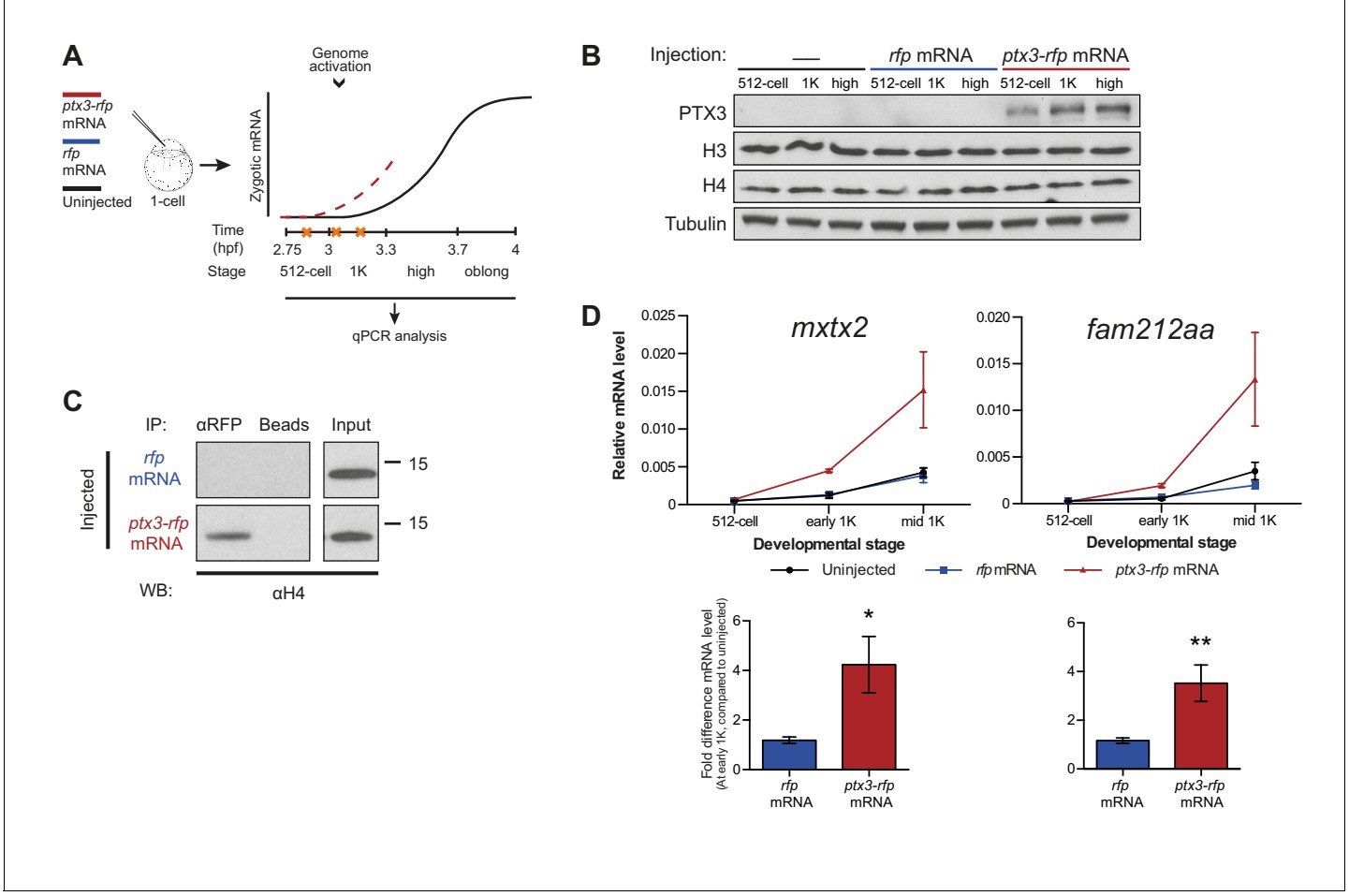

**Figure 3.** Decreasing the level of histones causes premature transcription. (**A**) Schematic representation of experimental procedure. *ptx3-rfp* or *rfp* (control) mRNA was injected into the cell of 1-cell embryos and qPCR analysis was carried out at stages around genome activation. Orange crosses represent the timing of stages used for the analysis. (**B**) Western blot analysis of PTX3, histone H3 and H4 levels at 512-cell, 1K and high stage in uninjected embryos, *rfp* and *ptx3-rfp* mRNA-injected embryos. Tubulin was used to control for equal loading. Blots shown are representative examples (n = 3). (**C**) Western blot analysis for histone H4 after a pull-down using an RFP antibody at 1K stage. Uncoupled beads were used as a negative control. Blot shown is a representative example (n = 3). (**D**) Expression of *mxtx2* and *fam212aa* was analyzed by qPCR at 512-cell, early 1K, and mid 1K stage in uninjected, *rfp* mRNA-injected and *ptx3-rfp* mRNA-injected embryos. Bar graphs focus on early 1K stage. Error bars represent SEM (n ≥ 4). *p<0.05; **p<0.01 (two-tailed Student's t-test, compared to *rfp* mRNA control). mRNA levels are normalized to the expression of *eif4g2α*.

The following figure supplement is available for figure 3:

**Figure supplement 1.** Decreasing the level of histones causes premature transcription.

example, at the 1-cell stage, there are 3098 times more histones per cell than are required to wrap the genome into chromatin. Due to the exponential increase in cell number during cleavage divisions, this number has dropped dramatically at 1K stage (**Figure 4C**). However, due to the large number of histones that is loaded in the oocyte, as well as the increase in histone level due to translation (**Figure 4B**), there are still nine genomes worth of non-DNA-bound histones per cell. Moreover, because what matters for protein-binding kinetics is the concentration, we next calculated the concentration of non-DNA-bound histones. Because the cleavage divisions are not accompanied by significant growth (the total animal cap volume increases by 29% from 128-cell to 1K, **Figure 4—figure supplement 1A**), the decreasing number of histones per cell is accompanied by a decreasing cellular volume, and the concentration of non-DNA bound histones in the cell does not change substantially (**Figure 4D**). Taken together, this shows that during transcription activation there is still a

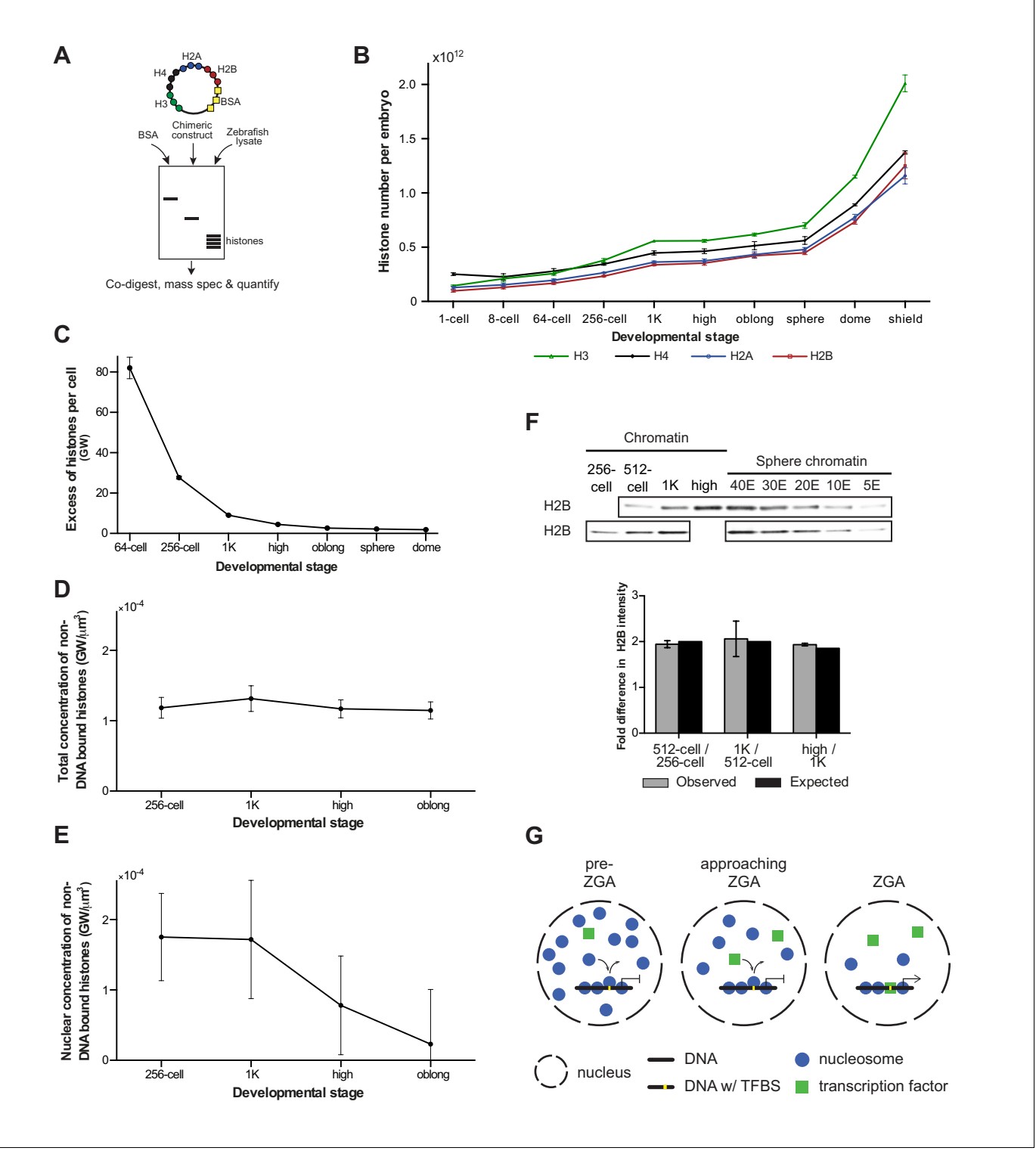

**Figure 4.** Onset of transcription coincides with a reduction in nuclear histone concentration. (**A**) Our quantitative mass spectrometry approach. Zebrafish histones were quantified by comparing the abundances of native histone peptides with corresponding isotopically labeled peptides from the chimeric protein; chimeric protein was quantified by comparing the abundance of labeled (from chimera) and native (from standard) BSA peptides (see Materials and methods for more details). (**B**) Quantification of the number of histone H3, H4, H2A, and H2B per embryo at indicated stages by quantitative mass spectrometry. Error bars represent SEM (n = 3). (**C**) The excess number of histones per cell (in genomes worth) was calculated using H2B levels (**Figure 4—source data 1**) and cell numbers (**Figure 1—source data 1**), and by assuming an average of 1.5 genomes per cell (see Materials and methods for more details). For better visualization of the data at later developmental stages the values for 1-cell and 8-cell are not shown in the

*Figure 4 continued on next page*

*Figure 4 continued*

graph but are 3098 and 518, respectively. Error bars represent SEM (n = 3). GW, genomes worth of histones. (**D**) The total concentration of non-DNA-bound histones was calculated by dividing the excess genomes worth of histone H2B per embryo by the volume of the animal cap at the respective stages (*Figure 4—figure supplement 1A*). Error bars represent SEM of animal cap volumes (n = 3). GW, genomes worth of histones. (**E**) The nuclear concentration of non-DNA-bound histones was calculated from immunofluorescence (from left to right n = 12, 12, 14, 15) combined with live imaging and mass spectrometry data (see Materials and methods for more details). Error bars represent SEM of animal cap volumes (n = 3). (**F**) Relative differences in H2B intensity between chromatin fractions of 256-cell, 512-cell, 1K, and high stage embryos. Sphere stage embryos were used to determine the linear range of H2B detection (see also *Figure 4—figure supplement 1C*). Blots shown are representative examples (n ≥ 3). Plots show observed fold differences in H2B intensity in chromatin fractions comparing indicated stages compared to the differences that would be expected if the intensity were to scale with the amount of DNA (E, embryo). (**G**) Competition model. See text for more details (TFBS, transcription-factor-binding site).

The following source data and figure supplement are available for figure 4:

**Source data 1.** Quantification of histone number by mass spectrometry.
**Source data 2.** Two channel recording of H4-sfGFP and PCNA-RFP distributions from 8-cell to oblong stage.
**Figure supplement 1.** Onset of transcription coincides with a reduction in nuclear histone concentration.

significant amount of non-DNA-bound histone and that the overall concentration of non-DNA-bound histones in the cell has not decreased by much.

Because transcription takes place in the nucleus, we next wanted to investigate the concentration of non-DNA-bound histones in this compartment of the cell. First, we analyzed the dynamics of histone localization by lightsheet microscopy of living embryos (*Figure 4—figure supplement 1B* and *Figure 4—source data 2*). As expected, we found a close coordination between the formation of nuclei after cell division and the import of histones from the cytoplasm into the nucleus: during each cell cycle, non-DNA-bound histones are concentrated in the nucleus. A direct quantification of non-DNA-bound, endogenous histones in the nucleus is difficult, but by combining lightsheet microscopy measurements of both the nuclear volume fraction and the relative fluorescence intensity of histone H4 in cytoplasm and nucleus with the absolute amount of histone H4 as quantified by mass spectrometry, we were able to calculate the nuclear concentration of non-DNA-bound histones from 256-cell to oblong stage (*Figure 4E*, *Figure 4—figure supplement 1A* and see Materials and methods for more details). Importantly, our calculations indicate a decrease in the nuclear concentration of non-DNA-bound histones at the onset of transcription. In combination with our finding that histone levels determine the timing of transcription, this suggests that a decrease in the concentration of non-DNA-bound histones in the nucleus causes the onset of transcription during embryogenesis.

## Nucleosome density on DNA is unchanged during genome activation

We next analyzed whether the decreased concentration of non-DNA bound histones in the nucleus is accompanied by a reduced density of nucleosomes on chromatin. We quantified the amount of histone H2B in the chromatin fraction of embryos ranging from 256-cell to high stage. Comparing the amount of histone H2B between stages revealed that the level of H2B scales with the amount of DNA (*Figure 4F* and *Figure 4—figure supplement 1C*). This is in agreement with a previous study in which we found that the density of nucleosomes does not significantly change during genome activation (*Zhang et al., 2014*). Our results reveal that global nucleosome density on DNA does not change during genome activation. Taken together, this suggests that the concentration of non-DNA-bound histones in the nucleus determines the timing of transcription without the need for a significant change in global nucleosome density.

## A competition model for the onset of transcription

Our finding that the concentration of histones in the nucleus determines the onset of transcription without a significant change in global nucleosome density on DNA suggests that a simple depletion model cannot explain a role for histone levels in the timing of zygotic transcription. To explain the effect of histone levels on zygotic genome activation, we hypothesized that the transcriptional machinery (for simplicity referred to as transcription factors) competes with nucleosome-forming

histones for binding to only a minimal fraction of the total DNA, corresponding to transcription-factor-binding sites (*Figure 4G*). In such a model, local substitution of nucleosomes by transcription factors allows for transcription to be activated, but will cause only localized changes in nucleosome positioning, and will barely affect the average nucleosome density. Transcription factors would lose the competition for DNA binding in the presence of an excess of histones (pre-ZGA), whereas a reduction of the concentration of non-DNA-bound histones in the nucleus would allow transcription factors to gain access to DNA (approaching ZGA) and initiate transcription (ZGA).

## Decreasing transcription factor levels delays the onset of transcription

If competition between nucleosome-forming histones and the transcriptional machinery determines the onset of transcription, it would be predicted that the levels at which transcription factors are present could also affect the timing of zygotic transcription. To test this, we changed the level of Pou5f3, a transcription factor that has been identified as being required for the activation of a large set of genes during genome activation (*Lee et al., 2013*; *Leichsenring et al., 2013*; *Onichtchouk et al., 2010*). To analyze the effect on the onset of transcription (*Figure 5A*), we selected five genes that are activated at the onset of genome activation and that have been identified as Pou5f3 targets (*Figure 1—figure supplement 1A*) (*Onichtchouk et al., 2010*).

To reduce the level of Pou5f3, we used a previously characterized morpholino (*Burgess et al., 2002*) and confirmed its effect by analyzing the morphology of injected embryos and the effect of the morpholino on the translation of injected RNA encoding Pou5f3 (*Figure 5—figure supplement 1A*). We verified that the selected Pou5f3-target genes require Pou5f3 for their expression and that other genes do not (*Figure 5—figure supplement 1B*), and analyzed the effect of a reduction in Pou5f3 levels on the timing of transcription of target genes (*Figure 5A*). Consistent with our model, a reduction in the amount of Pou5f3 delayed the onset of transcriptional activation: transcripts were detected in the middle of 1K stage in embryos injected with control morpholino, while in the embryos injected with Pou5f3 morpholino, the genes started to be transcribed at high stage (*Figure 5B* and *Figure 5—figure supplement 1C*). We analyzed embryos at mid 1K in this experiment, because the delay that we observe is weaker than with the histone cocktail. Comparison of gene expression levels in control morpholino and Pou5f3 morpholino-injected embryos at mid 1K (when transcripts can first be detected in control morpholino-injected embryos) revealed that the level of induction is significantly reduced upon injection of Pou5f3 morpholino (*Figure 5B* and *Figure 5—figure supplement 1C*, bar graphs). Performing similar experiments for two additional transcription factors (Sox19b and FoxH1) revealed that this effect is general, and not specific to Pou5f3 (*Figure 5—figure supplement 1D–G*). Staging by morphology was corroborated by cell counting with absolute time between the analyzed stages being constant (*Figure 5—figure supplement 1H*). Together, these data show that a decrease in the level of transcription factors in the embryo delays the onset of transcription of target genes.

## Increasing transcription factor levels causes premature transcription

Next, we analyzed the effect of increasing the level of Pou5f3 on the transcription of the selected Pou5f3 target genes (*Figure 5A*). We co-injected mRNA coding for Sox19b because it has been shown that Pou5f3 and Sox19b often co-occupy their target genes (*Chen et al., 2014*; *Leichsenring et al., 2013*; *Onichtchouk et al., 2010*). Injecting mRNA encoding these transcription factors resulted in overexpression of both proteins and the expected phenotypes for Pou5f3 overexpression (*Figure 5—figure supplement 2A*) (*Belting et al., 2011*). Although Pou5f3 was required for the expression of all genes we selected (*Figure 5—figure supplement 1B*), Pou5f3 and Sox19b were only sufficient to increase the expression level of *apoeb and dusp6* at high stage (*Figure 5—figure supplement 2B*). In agreement with this observation, overexpression of Pou5f3 and Sox19b resulted in premature expression of *apoeb and dusp6*: transcripts were detected at early 1K stage in embryos injected with *pouf53* and *sox19b*, whereas in the embryos injected with control mRNA, transcripts could be detected only at high stage (*Figure 5C*). Comparison of gene expression levels in control, and *pouf53* and *sox19b* mRNA-injected embryos at early 1K stage (one time-point prior to when genes are first induced in uninjected embryos) revealed that the level of expression is increased significantly upon injection of *pou5f3* and *sox19b* mRNA for *apoeb and dusp6* (*Figure 5C*, bar graphs). Such an effect on the timing of transcription was not observed for the other genes

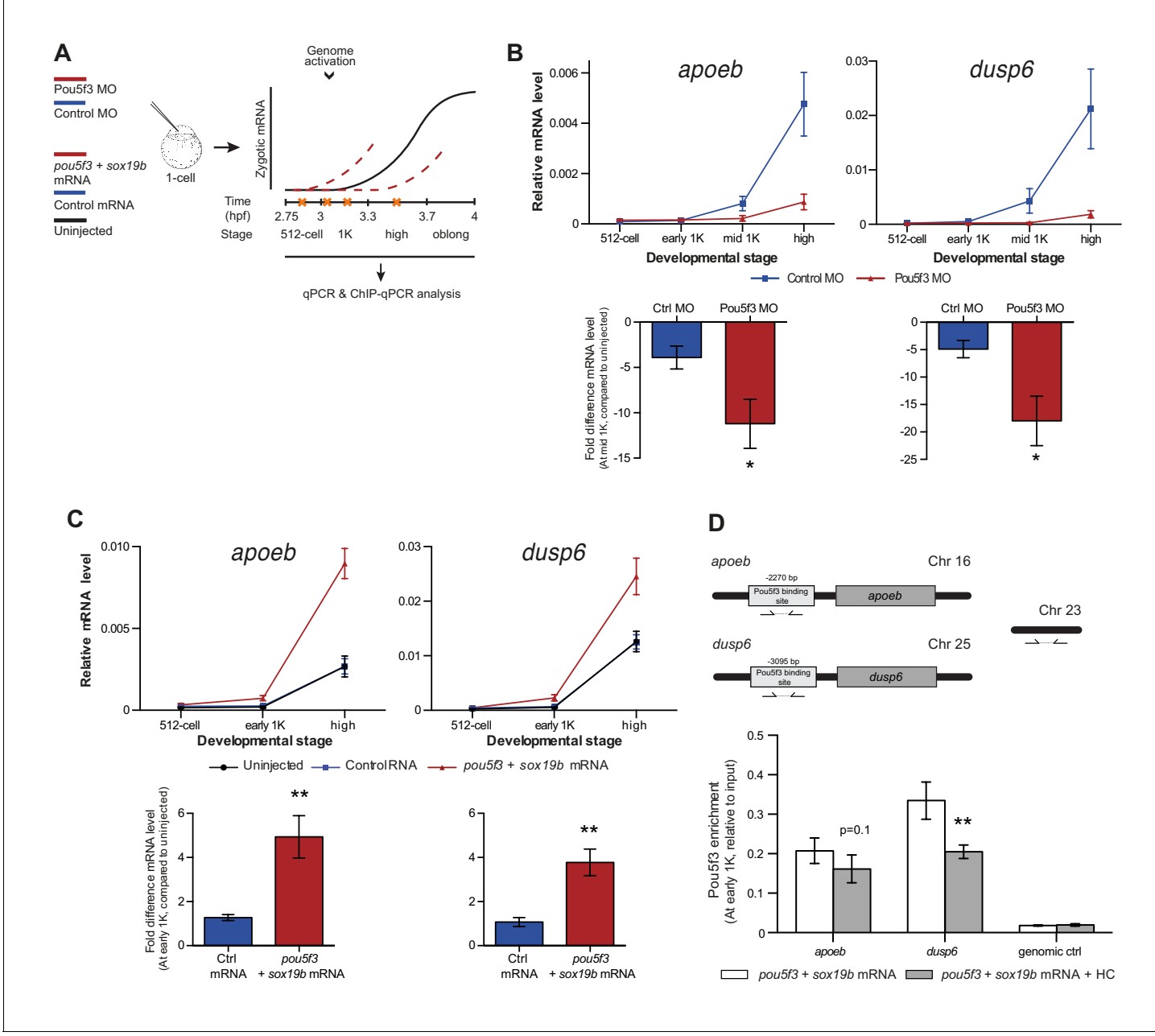

**Figure 5.** Direct experimental evidence for the competition model using endogenous genes and transcription factors. (**A**) Schematic representation of experimental procedure. Pou5f3 levels were decreased by injecting a morpholino, or increased by injecting *pou5f3* mRNA (in combination with *sox19b* mRNA) into the cell of 1-cell embryos. Controls used were a dead-end morpholino and *gfp* mRNA, respectively. qPCR and ChIP-qPCR analysis was carried out at stages around genome activation. Orange crosses represent the timing of stages used for the analysis. (**B**) Expression of *apoeb* and *dusp6* was analyzed by qPCR at 512-cell, early 1K, mid 1K and high stage in control and Pou5f3 morpholino-injected embryos. The data in the bar graphs focus on the mid 1K stage. Error bars represent SEM ($n \geq 4$). *$p<0.05$ (two-tailed Student's t-test, compared to control MO). (**C**) Expression of *apoeb* and *dusp6* was analyzed by qPCR at 512-cell, early 1K and high stage in uninjected embryos, embryos injected with control mRNA and embryos injected with *pou5f3* and *sox19b* mRNA. Bar graphs focus on the early 1K stage. Error bars represent SEM ($n \geq 4$). **$p<0.01$ (two-tailed Student's t-test, compared to control mRNA). (**D**) Binding of Pou5f3 to its respective binding sites for *apoeb* and *dusp6* (*Leichsenring et al., 2013*) and control region was analyzed by ChIP-qPCR at the early 1K stage in embryos injected with *pou5f3* + *sox19b* mRNA or *pou5f3* + *sox19b* mRNA plus histone cocktail. Enrichment of pulled-down fragments was normalized to input. Location of primer sets in respect to the transcription start-site used for ChIP-qPCR analysis are indicated by arrows. A genome control region on chromosome 23 was also used. Error bars represent SEM ($n = 5$). **$p<0.01$ (two-tailed Student's t-test ratio paired, compared to *pou5f3* + *sox19b* mRNA-injected embryos). In B and C, mRNA levels are normalized to the expression of *eif4g2α*.

*Figure 5 continued on next page*

*Figure 5 continued*

The following figure supplements are available for figure 5:

**Figure supplement 1.** Reducing transcription factor levels delays the onset of transcription.

**Figure supplement 2.** Increasing transcription factor levels causes premature transcription.

(*Figure 5—figure supplement 2C*). Staging by morphology was corroborated by cell counting, with absolute time between the analyzed stages being constant (*Figure 5—figure supplement 2D*). These experiments show that an increase in the level of Pou5f3 and Sox19b in the embryo can cause premature transcription. Taken together, our results show that changing the concentration of endogenous transcription factors can affect the onset of transcription. This is in agreement with our model in which the relative levels of histones and transcription factors determine the onset of transcription.

## Transcription factor binding is sensitive to histone levels

If transcription factors and histones compete for DNA binding, it would be predicted that transcription factor binding is sensitive to histone levels. To directly test competition at the chromatin level, we determined whether the binding of the transcriptional machinery is affected by histone levels. We analyzed the binding of Pou5f3 to its predicted target sites upstream of *apoeb* and *dusp6* by chromatin immunoprecipitation (ChIP) and identified co-precipitated DNA fragments by qPCR (*Figure 5D*). In embryos that were injected with mRNA encoding both Pou5f3 and Sox19b, we found that the binding of Pou5f3 was readily detected at early 1K stage (*Figure 5D*, white bars). When the HC was co-injected, binding of the transcription factor was reduced (*Figure 5D*, gray bars). We expect, but did not test, that nucleosome density is concordantly increased at these binding sites. A control region in genomic DNA did not show any binding of Pou5f3 (*Figure 5D*). Taken together, this shows that the binding of an endogenous transcription factor is sensitive to the amount of histones present in the embryo.

## Experimental evidence for competition using a heterologous transgene

Our results support a model in which transcription is regulated by the relative levels of histones and transcription factors. Endogenous gene regulation, however, is intrinsically complex, with multiple transcription factors providing input on the same gene, and often there is limited information on the number and strength of activator-binding sites. Because this might have affected the results we obtained with endogenous transcription factors and genes (*Figure 5*), we decided to take advantage of a heterologous system to confirm our results. The integrated inducible transgene TRE:GFP (*Figure 6A*) contains seven binding sites for tTA–VP16 as well as a CMV promoter and is strictly dependent on tTA–VP16 for its expression (data not shown). tTA-VP16 was tagged with HA and a protein product was detected at the 1K stage following injection of mRNA (*Figure 6—figure supplement 1A*). Injection of 5 pg of mRNA encoding the heterologous transcription factor tTA-VP16 resulted in the detection of transcripts at high stage, in accordance with the onset of zygotic transcription of endogenous genes (*Figure 6B*). Next, we analyzed the transcriptional activity of this transgene upon injection of 300 pg of mRNA encoding tTA-VP16 and we observed that transcripts could be detected at early 1K (*Figure 6B*). Comparison of gene expression levels at early 1K stage (one stage prior to when genes are first induced in embryos injected with 5 pg of mRNA) in embryos injected with 5 and 300 pg of mRNA revealed that the number of transcripts is increased significantly upon increasing the level of transcription factor (*Figure 6B*, bar graph). Next, we tested whether an increase in histone levels would negate the effect of high levels of transcription factor. As predicted by the competition model, the increase in transcriptional activity that is observed upon the injection of 300 pg of tTA-VP16 mRNA is lost when the histone cocktail is co-injected (*Figure 6C*). Finally, we determined whether the binding of tTA is affected by histone levels. We analyzed the binding of tTA-VP16 to the TRE sites in the transgene by ChIP-qPCR (*Figure 6A*). We found that upon injecting 300 pg of tTA-VP16 mRNA, the binding of the transcription factor was readily detected at early 1K stage (*Figure 6D* and *Figure 6—figure supplement 1B*). As expected, binding of the transcription factor was significantly reduced when the HC was co-injected. Control regions within the transgene

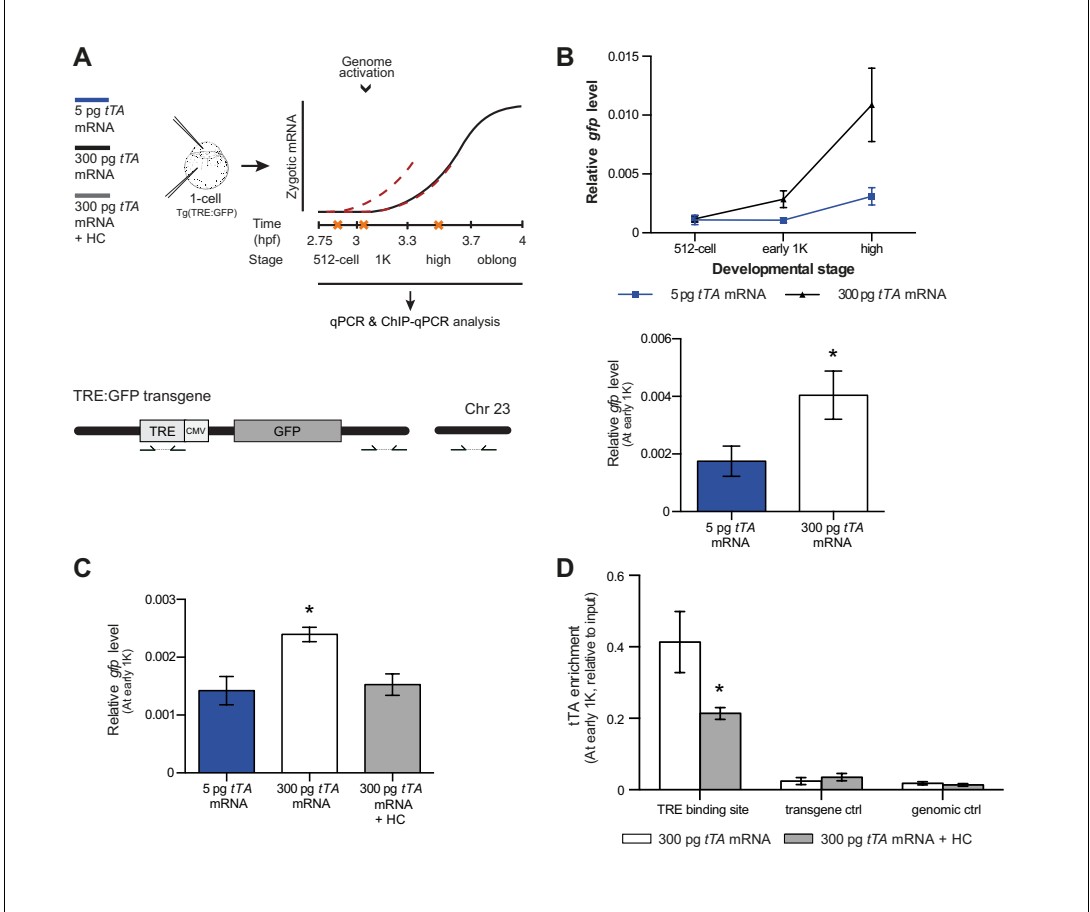

**Figure 6.** Direct experimental evidence for the competition model using a heterologous transgene. (**A**) Schematic representation of the experimental procedure and TRE:GFP transgene. tTA-VP16 and/or histone levels were increased by injecting mRNA or HC into the cell or yolk, respectively, of 1-cell transgenic embryos. The TRE element contains seven binding sites for tTA-VP16 and is joined to a CMV promoter. qPCR and ChIP-qPCR analysis was carried out at stages around genome activation. Orange crosses represent the timing of stages used for the analysis. (**B**) Expression of *gfp* was analyzed by qPCR at 512-cell, early 1K and high stage in embryos injected with 5 or 300 pg *tTA-VP16* mRNA. Bar graphs focus on the early 1K stage. Error bars represent SEM (n ≥ 4). *p<0.05 (two-tailed Student's t-test, compared to 5 pg *tTA-VP16* mRNA). (**C**) Expression of *gfp* was analyzed by qPCR at early 1K stage in embryos injected with 5 pg, 300 pg *tTA-VP16* mRNA and 300 pg *tTA-VP16* mRNA plus histone cocktail. Error bars represent SEM (n = 4). *p<0.05 (Ordinary one-way ANOVA). (**D**) Binding of tTA-VP16 to the TRE element and control regions was analyzed by ChIP-qPCR at the early 1K stage in embryos injected with 300 pg *tTA-VP16* mRNA or 300 pg *tTA-VP16* mRNA plus histone cocktail. Enrichment of pulled-down fragments was normalized to input. Primer sets used for ChIP-qPCR analysis are indicated by arrows in panel A. A control region on the transgene was used in addition to a genome control region on chromosome 23. Error bars represent SEM (n = 3). *p<0.05 (two-tailed Student's t-test ratio paired, compared to 300 pg *tTA-VP16* mRNA-injected embryos). In B and C, mRNA levels are normalized to the expression of *eif4g2α*.

The following figure supplement is available for figure 6:

**Figure supplement 1.** Direct experimental evidence for the competition model using a heterologous transgene.

and in genomic DNA did not show any binding of tTA-VP16 (*Figure 6D* and *Figure 6—figure supplement 1B*). This shows that the binding of a heterologous transcription factor is sensitive to the amount of histones present in the embryo. These results are in agreement with the results obtained with the endogenous transcription factors (*Figure 5*). Taken together, our data provide direct evidence for a model in which competition between histones and transcription factors determines the onset of transcription.

## Discussion

In this study, we have shown that the concentration of all four core histones determines the onset of transcription in zebrafish embryos by competing with transcription factors for binding to DNA. Upon fertilization, there is a large excess of histones stockpiled in the embryo and transcription starts when the concentration of non-DNA-bound histones in the nucleus drops, and the transcriptional machinery gains access to DNA. Thus, the relative concentrations of both histones and transcription factors determine the timing of zygotic genome activation (*Figure 4G*). Our observations provide, to our knowledge, the first example of a developmental transition in which competition for DNA binding between histones and transcription factors plays an important role in transcriptional regulation.

### All core histones are important for timing of transcription

Our observation that histone levels affect the time of transcription onset in zebrafish embryos is in agreement with previous studies that showed a role of histones in the regulation of transcription in early *Xenopus* embryos and extracts (*Almouzni and Wolffe, 1995*; *Amodeo et al., 2015*). However, our finding that histones are neither completely depleted from the soluble fraction, nor generally depleted from chromatin, argues against a model in which a global loss of nucleosome density on chromatin causes the onset of transcription (*Amodeo et al., 2015*). Our work does not exclude the importance of other factors, such as the linker histone H1 (*Pérez-Montero et al., 2013*), the embryonic form of which is stably present during zebrafish genome activation, but it establishes that core histones themselves function as actual repressors of transcription.

Our discovery that all core histones are required to regulate the onset of transcription suggests that the nucleosome is important for the repressor function of histones. In a previous study, premature transcription of injected plasmids caused by an excess of non-specific DNA was negated by the addition of the four core histones, but histones were not tested separately and it was not clear whether one or more histones were required for the observed effect on transcription (*Almouzni and Wolffe, 1995*). This left open the possibility that single histones could repress transcriptional activity in the embryo, for example by binding to a transcription factor and preventing it from binding to DNA. However, our observation that in the embryo all core histones are important for the regulation of transcription would then require all histones to independently take part in this mode of repression. Because this is a very unlikely scenario, we propose that repression takes place close to DNA, where histones are assembled into a histone octamer to form the nucleosome.

### Changes in nuclear histone concentration during genome activation

We propose that genome activation follows a decrease in the concentration of non-DNA bound histones in the nucleus. One possible way to explain the reduction in nuclear histone concentration is the exponential increase in DNA content during the cleavage stages of zebrafish development. Because histones have a high affinity for DNA, an increase in the amount of DNA would titrate out non-DNA-bound histones. Several experiments have indeed shown that changes in DNA content can affect the time of transcription in *Drosophila*, zebrafish and *Xenopus* (*Dekens et al., 2003*; *Lu et al., 2009*; *Newport and Kirschner, 1982a*; *Prioleau et al., 1994*). Our data suggest that these effects were the result of reducing the concentration of non-DNA-bound histones. In zebrafish embryos, the amount of histones is so large that the increase in DNA content leading up to genome activation may contribute only moderately to a decrease in the concentration of non-DNA-bound histones in the nucleus.

Another possible explanation for the decrease in nuclear histone concentration is the marked increase in the ratio of nuclear over cytoplasmic volume during the cleavage stages (*Figure 4—figure supplement 1A*). We suggest that this may limit the capacity of the nucleus to concentrate histones. The process of nuclear transport has been investigated in great detail (*Kim and Elbaum, 2013a*, *2013b*; *Kopito and Elbaum, 2007*, *2009*), and we can assume that the nuclear envelope can create a certain fold difference in concentrations between the nucleus and cytoplasm. During the initial stages of zebrafish development, the nucleus occupies only a very small fraction of the cell volume (1.1% at 128-cell stage, *Figure 4—figure supplement 1A*). As a result, when the nucleus concentrates histones up to the maximum fold difference between cytoplasm and nucleus, the cytoplasmic histone concentration is hardly altered. Later in development, when approaching the onset of zygotic transcription, the nucleus takes up a notably larger part of the total cell volume (7.1% at

high stage, *Figure 4—figure supplement 1A*). Now, when histones are imported into the nucleus, the concentration of histones in the cytoplasm noticeably decreases. The nuclear envelope is still able to create approximately the same fold difference of concentrations, but due to the reduced cytoplasmic concentration, the achieved nuclear concentration is not as high as during the initial stages. Thus, in this scenario, the nuclear histone concentration decreases due to the increasing relative nuclear size, which alters the distribution of histones among cellular compartments. It may be expected that the relative increase in nuclear size affects the nuclear concentration of transcription factors as well, and it remains to be seen how the concentration of histones and transcription factors change with respect to each other in order to activate transcription. Experiments recently performed in *Xenopus* showed that changing the size of the nucleus affects the timing of transcription (*Jevtić and Levy, 2015*), providing further evidence for a role of nuclear size in regulating the onset of transcription.

## Competition for DNA binding determines transcription onset

Our finding that the onset of transcription depends on the concentration of histones, but also on the concentration of transcription factors, is consistent with previous studies that suggested an important role for transcriptional activators in the temporal regulation of zygotic transcription (*Almouzni and Wolffe, 1995*; *Prioleau et al., 1994*; *Veenstra et al., 1999*). Because transcription can be induced prior to the onset of genome activation, both by adding DNA (*Dekens et al., 2003*; *Lu et al., 2009*; *Newport and Kirschner, 1982a*) or removing histones (this study), we suggest that transcription factors required for the onset of transcription are in principle present prior to genome activation. To shift the balance from repression to activation, the relative concentrations of histones and transcription factors need to be changed in favor of transcription factors. This would explain previous observations in *Xenopus*, where the addition of TBP (in combination with adding DNA) or GAL4-VP16 resulted in premature transcription (*Almouzni and Wolffe, 1995*; *Veenstra et al., 1999*). Based on our findings, those experiments would have shifted the balance in favor of transcriptional activity, similar to the effect observed when we increased transcription factor levels or decreased histone levels.

Our model in which histones and transcription factors dynamically compete for DNA binding to regulate transcription in the embryo is consistent with the notion that most transcription factors cannot bind DNA when it is wrapped around a nucleosome and thus compete with nucleosomes for DNA access (*Almouzni and Wolffe, 1993*; *Almouzni et al., 1990*; *Hayes and Wolffe, 1992*; *Miller and Widom, 2003*; *Mirny, 2010*; *Ramachandran and Henikoff, 2016*; *Raveh-Sadka et al., 2012*; *Schild-Poulter et al., 1996*; *Svaren et al., 1994*). Initially, there is a large excess of histones stockpiled in the embryo and transcription starts when the concentration of non-DNA-bound histones in the nucleus drops and the transcriptional machinery gains access to DNA. In contrast to a competition model that was previously proposed (*Prioleau et al., 1994*, *1995*), transcription factors do not need to be pre-bound to DNA in order to compete with histones, but rather, they dynamically compete with histones for DNA binding.

Our experiments did not address when competition takes place during the cell cycle. Competition for DNA binding might either occur immediately following replication, on temporarily naked DNA, or following chromatin assembly. Recent studies assessing the nucleosome landscape following replication have revealed that replication-coupled nucleosome assembly initially outcompetes transcription factors for binding to DNA but that chromatin remodeling and phasing of nucleosomes by remodelers and transcription factors occurs rapidly thereafter (*Fennessy and Owen-Hughes, 2016*; *Ramachandran and Henikoff, 2016*; *Vasseur et al., 2016*). Future experiments will determine the details of competition in the early embryo, with its rapid cell cycles and large pool of soluble histones.

To gain further insight in the molecular details of competition that lead up to genome activation, it will be important to determine which factors compete with histones for DNA binding. In theory, the binding of all factors that require access to DNA could be affected by histone levels, suggesting that competition might take place at many levels of transcription regulation: the formation of higher order chromatin structure, chromatin remodeling, the binding of transcription factors, and the assembly of the basal transcription complex. Our results show that the transcription factors that have been identified to regulate many genes during genome activation in zebrafish (Pou5f3 and Sox19b) (*Lee et al., 2013*; *Leichsenring et al., 2013*), as well as FoxH1 and the heterologous

transcription factor tTA-VP16, compete with histones for binding to DNA (*Figures 5* and *6*). In this context, it is interesting to note that the transcription factors that have been identified to play a role in genome activation have either been suggested to be pioneer factors (*Lee et al., 2013*; *Leichsenring et al., 2013*), or there is indication for such a role because of their homology with mammalian pioneer factors (*Lee et al., 2013*; *Leichsenring et al., 2013*; *Soufi et al., 2012*). Pioneer factors are able to interact with DNA that is nucleosome bound (*Zaret and Carroll, 2011*). In the context of the competition model, it will be interesting to see whether these factors also have pioneering activity in the early embryo, and how this affects their role in activating transcription in the embryo.

## General relevance of competition in development

The applicability of the competition model might extend well beyond the onset of zygotic transcription in zebrafish. First, given the excess of histones in a large number of species including *Drosophila*, *Xenopus*, and zebrafish (*Adamson and Woodland, 1974*; *Li et al., 2012*; *Marzluff and Duronio, 2002*; *Osley, 1991*; *Vastenhouw et al., 2010*; *Woodland and Adamson, 1977*), it is likely that histone levels play a role in the timing of zygotic transcription across these species. As discussed, in *Xenopus* embryos there is indeed evidence for a role of histone levels in regulating transcriptional activity (*Almouzni and Wolffe, 1995*; *Amodeo et al., 2015*). Additional experiments will be required to determine whether the competition model we propose applies to these and other species. Second, the onset of zygotic transcription in the embryo takes place in the context of the mid-blastula transition and is accompanied by a lengthening of the cell cycle and changes in chromatin structure. Although it had previously been suggested that the rapid cell cycles lacking G1 and G2 phases might interfere with productive transcription during early developmental stages (*Collart et al., 2013*; *Edgar and Schubiger, 1986*; *Kimelman et al., 1987*), it was recently shown that the lengthening of the cell cycle might be a direct consequence of the onset of transcription in *Drosophila* embryos (*Blythe and Wieschaus, 2015b*). This would suggest that what regulates the onset of zygotic transcription might also dictate the lengthening of the cell cycle. Finally, post-translational modification of histones often requires a chromatin-modifying enzyme to bind to DNA, much like transcription factors. Thus, competition is likely to affect the de novo modification of histones as well, explaining why many histone modifications are only observed around the onset of zygotic transcription in their temporal profile (*Lindeman et al., 2011*; *Vastenhouw et al., 2010*). Importantly, we observe an effect on the timing of transcription by adding unmodified histones. This suggests that post-translational modifications of histones are either downstream of the timing of transcriptional activation, or the enzymes that modify histones are not limiting in the embryo.

The competition model can explain why genome activation is gene specific (*Aanes et al., 2011*; *Collart et al., 2014*; *Harvey et al., 2013*; *Heyn et al., 2014*; *Lott et al., 2011*; *Owens et al., 2016*; *Pauli et al., 2012*; *Sandler and Stathopoulos, 2016*; *Tan et al., 2013*), and even why the first zygotic transcripts can be detected several cell cycles before the stage that is traditionally defined as the time point of ZGA (*De Renzis et al., 2007*; *Heyn et al., 2014*; *Skirkanich et al., 2011*; *Yang et al., 2002*). The sensitivity of genes for a given histone concentration logically depends on their enhancers and the affinity and concentration of the transcription factors that bind to them. In this context, it is interesting to note that many genes that are activated during zebrafish genome activation respond to the same set of transcription factors, which are also the most highly translated transcription factors before genome activation (*Lee et al., 2013*; *Leichsenring et al., 2013*). Conversely, the affinity of transcription factors and the number of transcription-factor-binding sites might provide a mechanism to explain why some genes overcome repression earlier than others (*Heyn et al., 2014*). Indeed, in *Drosophila*, it has been shown that the number of transcription-factor-binding sites as well as the level of transcription factors can affect the timing of gene expression (*Foo et al., 2014*; *Harrison et al., 2010*).

Recent literature suggests that histone levels might play a role in the regulation of transcription during developmental transitions other than genome activation. In contrast to the situation in early embryos, where histone and DNA levels do not scale, it was generally believed that in somatic cells, histone production is tightly coupled to replication (*Nurse, 1983*). Recently, however, histone levels have been shown to change during ageing and differentiation (*Feser et al., 2010*; *Hu et al., 2014*; *Karnavas et al., 2014*; *O'Sullivan et al., 2010*) suggesting that histone levels might not be as tightly coupled to DNA replication as previously thought. Moreover, histone chaperones were identified as

inhibitors of reprogramming (*Cheloufi et al., 2015*; *Ishiuchi et al., 2015*) and it was proposed that the lack of histone chaperones facilitates transcription factor binding (*Cheloufi et al., 2015*). Taken together, these studies might suggest that the availability of histones could play a role in the regulation of transcription during differentiation and reprogramming.

We have shown that the onset of transcription is regulated by a dynamic competition for DNA binding between histones and transcription factors. This suggests that the relative levels of histones and transcription factors in the nucleus determine the time at which transcription begins in the embryo. Future studies will be required to improve our understanding of the molecular mechanism of competition, the regulation of repressor and activator concentrations in the nucleus, and the role of competition during other developmental transitions.

## Materials and methods

### Zebrafish husbandry and manipulation

Zebrafish were maintained and raised under standard conditions. Wild-type (TLAB) (WT-TL RRID: ZIRC_ZL86, WT-AB RRID:ZIRC_ZL1) and transgenic embryos were dechorionated immediately upon fertilization, synchronized and allowed to develop to the desired stage at 28°C. Stage was determined by morphology and corroborated by cell counting. In terms of absolute time, the time between collected stages around ZGA was consistent between all conditions within an experiment and for all experiments. Histone cocktail and BSA (A9418; Sigma, St. Louis, MO) were injected into the yolk at the 1-cell stage at 22 ng per embryo. Pou5f3 anti-sense morpholino was injected at 6 ng per embryo, together with 1 ng of p53 morpholino (*Langheinrich et al., 2002*). Sox19b anti-sense morpholino (*Okuda et al., 2010*) was injected at 2 ng per embryo and FoxH1 anti-sense morpholino (*Pei et al., 2007*) was injected at 4 ng per embryo, together with 1 ng of p53 morpholino. Dead-end (*Weidinger et al., 2003*) or control morpholino were injected as a control at the same concentration. Morpholino sequences can be found in *Table 1*. α amanitin (A2263; Sigma) was injected at the 1-cell stage at a concentration of 0.2 ng per embryo. 2.8 mg/ml rhodamine-dextran (D3307; Molecular Probes, Eugene, OR) was used as an injection marker for the HC and BSA experiments. For all other injections, 0.1% Phenol red (P0290; Sigma) was injected. Bright-field images of whole embryos were acquired on a Leica M165 C dissecting scope equipped with a Leica MC170 HD camera (Leica, Wetzlar, Germany).

### mRNA production and injection

mRNA was synthesized using the Ambion mMESSAGE mMACHINE SP6 Transcription Kit (AM1430; ThermoFisher Scientific, Waltham, MA). Human PTX3 cDNA was cloned into a pCS2+ vector with a C-terminal RFP. *ptx3-rfp* and *rfp* mRNA were injected into the cell at the 1-cell stage at a concentration of 300 pg per embryo. Zebrafish Pou5f3 and Sox19b cDNA were cloned into a pCS2+ vector containing 2xHA sequences. For gene expression experiments, *pou5f3-2xHA* and *sox19b-2xHA* mRNA were each injected into the cell of 1-cell embryos at 300 pg per embryo. mRNA encoding cytoplasmic *gfp* was injected as a control at 600 pg per embryo. For ChIP-qPCR experiments, *pou5f3-2xHA* and *sox19b-mEos2* mRNA were each injected into the cell of 1 cell embryos at 150 pg

**Table 1.** List of morpholinos used.

**Morpholinos**

| Target | Sequence | Company | Reference |
|---|---|---|---|
| p53 | 5'-GCGCCATTGCTTTGCAAGAATTG | GeneTools | *Langheinrich et al. (2002)* |
| Pou5f3 | 5'-CGCTCTCTCCGTCATCTTTCCGCTA | GeneTools | *Burgess et al. (2002)* |
| Sox19b | 5'-ACGAGCGAGCCTAATCAGGTCAAAC | GeneTools | *Okuda et al. (2010)* |
| Foxh1 | 5'-TGCTTTGTCATGCTGATGTAGTGGG | GeneTools | *Pei et al. (2007)* |
| Dead-end | 5'-GCTGGGCATCCATGTCTCCGACCAT | GeneTools | *Weidinger et al. (2003)* |
| Ctrl MO | 5'-CCTCTTACCTCAGTTACAATTTATA | GeneTools | GeneTools, LLC |

per embryo. A subsequent injection of either histone cocktail or mock (histone buffer) into the yolk was carried out. Human H4 cDNA was cloned into a pCS2+ vector with C-terminal sfGFP (50550; addgene, Cambridge, MA) (*Olson et al., 2014*). mRNA encoding H4-sfGFP was injected into the cell of 1-cell embryos at 240 pg per embryo. tTA-VP16 DNA was cloned into a pCS2+ vector containing 2xHA sequences. mRNA encoding tTA-VP16-2xHA was injected into the cell of 1-cell Tg (TRE:GFP) embryos either at 5 pg or 300 pg per embryo. The combination injection of 300 pg *tTA-VP16-2xHA* mRNA and histone cocktail, involved two subsequent injections into the cell of 1-cell embryos and yolk, respectively. The 300 pg *tTA-VP16-2xHA* mRNA only injections also received a secondary mock injection into the yolk.

## Quantitative PCR

Twenty-five embryos per developmental stage were snap frozen in liquid nitrogen. RNeasy Mini Kit (74104; Qiagen, Venlo, the Netherlands) was used to extract RNA. For Tg(TRE:GFP) embryos, contaminating DNA was removed from RNA preparations using the DNA-free Kit (AM1906; Thermo-Fisher Scientific). mRNA was converted to cDNA using the iScript cDNA Synthesis Kit (1708891; Bio-Rad Laboratories, Hercules, CA). SYBR green (AB-1158.; ThermoFisher Scientific) with Rox (R1371; ThermoFisher Scientific; 100 nM) was used as the qPCR master mix. Primers were used at a final concentration of 500 nM and sequences can be found in *Table 2*. Two or three technical replicates were performed for each sample. Ct values were normalized to the maternally loaded gene *eif4g2a* or input in ChIP-qPCR analysis. Relative mRNA expression levels were calculated via $1/(2^{(gene-eif4g2a)})$. Fold difference was calculated by dividing the relative mRNA expression level value of the test sample over control.

## Staging embryos by cell counting

Embryos were fixed in 4% formaldehyde in Danieau's solution at 4°C overnight. The next day, embryos were washed with Danieau's solution and then permeabilized with 0.2% Triton X in Danieau's solution for 30 min. Subsequently, embryos were incubated for 10 min in DAPI (1 µg/ml) and then washed several times with Danieau's solution. Embryos were placed in an inverted agarose holder and covered with Danieau's solution for imaging. An upright Zeiss LSM 780 NLO microscope equipped with a coherent Chameleon Vision II infrared laser was used for two-photon excitation (Carl Zeiss AG, Oberkochen, Germany). DAPI was excited with 780 nm and detected using a non-descanned GaAsP detector (BIG-Module) with BP450/60 or SP485. Samples were imaged with either a Zeiss W Plan-Apochromat 20 × 1.0 or 40 × 1.0 dipping objective. Images were acquired using a four tile scan of multiple z-sections (3–3.5 µm steps). Tiles were stitched with the ZEN software (RRID:SCR_013672, Zeiss). Images were imported into the Imaris software (RRID:SCR_007370, Bitplane, Belfast, Northern Ireland) and the spot tool was used to calculate cell number.

## Western blotting

Embryos were manually deyolked at the desired stage and snap frozen in liquid nitrogen. For all proteins, equal numbers of embryos were analyzed for each developmental stage (H4 and H2A [$n = 10$], all other proteins [$n = 5$]). Samples were boiled with SDS loading buffer at 98°C, run on 4–12% polyacrylamide NuPAGE Bis-Tris gels (NP0321BOX; ThermoFisher Scientific) and blotted onto a nitrocellulose membrane (10600002; GE Life Sciences, Chicago, IL). Primary antibodies were incubated at RT for 1 hr or overnight at 4°C and secondary antibodies were incubated at RT for 45 min. Primary and secondary antibodies used are listed in *Tables 3* and *4*, respectively. Membranes were analyzed on an Odyssey Infrared Imaging System (LI-COR, Lincoln, NE) or via chemiluminescent detection (GE Life Sciences) and X-ray film (GE Life Sciences). Tubulin was examined visually on all blots as a loading control.

## Quantitative mass spectrometry

We selected five proteotypic peptides (*Worboys et al., 2014*) for each of the four histones: H3, H4, H2A and H2B. The peptides do not discriminate between known histone variants for H3 and H2A. A chimeric gene encoding these peptides (*Beynon et al., 2005*) in addition to reference peptides from BSA and Glycjogen Phosphorylase B (PhosB) (five each), and flanked by Strep- and His-tags, was chemically synthesized (Gene Art, ThermoFisher Scientific). This gene was expressed in a Lys,

**Table 2.** List of primers used. Location of primer sets with respect to transcription start-sites are indicated in brackets.

**Primer list**

| Gene | Primers |
|---|---|
| eif4g2α | 5'-GAGATGTATGCCACTGATGAT |
| | 5'-GCGCAGTAACATTCCTTTAG |
| mxtx2 | 5'-ACTGACTGCATTGCTCAA |
| | 5'-ACCATACCTGAATACGTGATT |
| fam212aa | 5'-GCAAATGAGTATCTAAAACTGCT |
| | 5'-CATCATATAGCGCATCTGGT |
| nnr | 5'-GAGACATACCACAGGTGAAGC |
| | 5'-CCGCTCTGGTCTGTTGC |
| vox | 5'-TTATTCGTCGGGTTATGAGAG |
| | 5'-AACCAAGTTCTGATCTGTGT |
| sox19a | 5'-GAGGATGGACAGCTACGG |
| | 5'-CTATAGGACATGGGGTTGTAG |
| grhl3 | 5'-AGACGAGCAGAGAGTCCT |
| | 5'-TTGCTGTAATGCTCGATGATG |
| apoeb | 5'-GCAGAGAGCTTGACACACTAA |
| | 5'-TGCATTCTGCTCCATCATGG |
| dusp6 | 5'-AGCCATCAGCTTTATTGATGAG |
| | 5'-CAAAGTCCAAGAGTTGACCC |
| klf17 | 5'-ATAGTTCGGGACTGGAAAGTTG |
| | 5'- TGAGGTGTTGTCGTTGTCAG |
| irx7 | 5'-TGGCACACATTAGCAATTCC |
| | 5'-GCATGATCTTCTCGCCTTTG |
| klf2b | 5'-GCTCTGGGAGGATAGATGGA |
| | 5'-CTCGGAGTGGGAGATGAAC |
| flh | 5'-CACTGAAGCTCAGGTTAAAGTC |
| | 5'-ACAATCTGGGGAAAATCATGG |
| wnt11 | 5'-CAGACAGGTGCTTATGGACT |
| | 5'-CATCTCTCGGGGCACAAG |
| gadd45bb | 5'-CAACTCATGAATGTGGATCCAG |
| | 5'-ATGCAGTGAAGGTCTCTTGG |
| GFP | 5'-GCACCATCTTCTTCAAGGAC |
| | 5'-TTGTCGGCCATGATATAGAC |
| Pou5f3 binding site (−2270 apoeb) | 5'-TAAAGTGAGCAAATGTATGGCC |
| | 5'-TTTGTTGATTAAATCGCTTGTGA |
| Pou5f3 binding site (−3095 dusp6) | 5'-CATATGTTAAGCGGGGTGAAAC |
| | 5'-ATCCTGTCTCCTGTGTCATTTG |
| TRE binding site (−222) | 5'-TCTTGATAGAGAGGCTGCAAAT |
| | 5'-TCGAGATGGGCCCTTGATA |
| TRE binding site (13) | 5'-TCGTATAGGGATAACAGGGTAATG |
| | 5'-TACACGCCTACCTCGACC |
| TRE binding site (217) | 5'-GTACGGTGGGAGGCCTATAT |
| | 5'-CTTCTATGGAGGTCAAAACAGC |

*Table 2 continued on next page*

*Table 2 continued*

**Primer list**

| Gene | Primers |
|---|---|
| Transgene control | 5'-CTCTACAAATGTGGTATGGCTG |
| | 5'-ATTACCCTGTTATCCCTAAGGC |
| Genomic control | 5'-CCATCATATTCACATCTTGCAAG |
| | 5'-GTTCGTATGAACCGGAAGC |

Arg dual-auxotroph *E. coli* strain (BL21DE3pRARE) that was grown in media complemented with $^{13}C^{15}N$-Arg and $^{13}C$-Lys (Silantes, Munich, Germany). In a separate LC-MS/MS experiment, we established that the full-length chimeric protein was correctly expressed and the rate of incorporation of isotopically labeled amino acids was ca. 99%. The gel band corresponding to the chimeric protein was co-digested with the gel slab containing the histones from the samples of interest (*Shevchenko et al., 2006*) and with the band containing the exactly known amount of the reference protein (BSA). The recovered tryptic peptides were analysed by nanoLC-MS/MS on a LTQ Orbitrap Velos coupled with Dionex Ultimate 3000 nano-HPLC system (ThermoFisher Scientific). Three biological replicates were analyzed for each sample, with two technical replicates for each sample. The peptides were separated using C18 reversed phase column (Acclaim PepMap 100) over a linear gradient from 0 to 55% solvent B in a mixture of solvents A and B, delivered in 120 min (Solvent A 0.1% FA, Solvent B 60% ACN + 0.1% FA). The identification of peptides was performed using Mascot v2.2.04 (Matrix Science, London, United Kingdom) against a custom-made database composed of sequences from all histones, BSA, PhosB, affinity tag and the sequences of common contaminants such as human keratins and porcine trypsin. The raw abundances of extracted ion chromatograms (XIC) peaks of peptide precursors were reported by Progenesis LC-MS v4.1 software (Nonlinear Dynamics, Newcastle, United Kingdom). First the chimeric protein was quantified by comparing the abundances of BSA peptides comprised in its sequence with the corresponding peptides obtained by co-digestion of a known amount of BSA protein standard. In turn, the molar content of target zebrafish histones was inferred from the content of the chimera protein and the ratio of relative abundances of XIC peaks of precursor ions of matching pairs of labeled (originating from chimera) and unlabeled (originating from histones) peptides. Note that all peptides were recovered from the same in-gel digest and quantified at the same LC-MS/MS run.

**Table 3.** List of primary antibodies used.

**Primary antibodies**

| Target/Name | Company | Company code | RRID | [WB] | [IF] | [IP] |
|---|---|---|---|---|---|---|
| H3 | Abcam | ab1791 | AB_302613 | 1:10,000 | | |
| H4 | Abcam | ab10158 | AB_296888 | 1:1000 | 1:300 | |
| H2A | Abcam | ab18255 | AB_470265 | 1:1000 | | |
| H2B | Abcam | ab1790 | AB_302612 | 1:3000 | | |
| α-tubulin | Sigma | T6074 | AB_477582 | 1:20,000 | | |
| RFP | Abcam | ab152123 | AB_2637080 | | | Excess |
| PTX3 | Cosmo Bio | PPZ1724 | AB_1962280 | 1:15,000 | | |
| RNA Pol II | BioLegend | MMS-126R | AB_10013665 | | 1:1000 | |
| HA | Abcam | ab9110 | AB_307019 | 1:5000 | | Excess |
| IgG from rabbit serum | Sigma | I5006 | AB_1163659 | | | Excess |

WB, Western blotting; IF, immunofluorescence; IP, immunoprecipitation.

**Table 4.** List of secondary antibodies used.

**Secondary antibodies**

| Name | Company | Company code | RRID | [WB] | [IF] |
|------|---------|--------------|------|------|------|
| Alexa 488 goat anti-mouse IgG H&L | ThermoFisher | A-11029 | AB_138404 | | 1:1000 |
| Alexa 594 goat anti-rabbit IgG H&L | ThermoFisher | A-11037 | AB_2534095 | | 1:500 |
| IRDye 800CW donkey anti-rabbit IgG H&L | LI-COR | P/N 926–32213 | AB_621848 | 1:20,000 | |
| IRDye 800CW donkey anti-mouse IgG H&L | LI-COR | P/N 926–32212 | AB_621847 | 1:20,000 | |
| Peroxidase AffiniPure goat anti-rabbit IgG H&L | Jackson ImmunoResearch | 111-035-144 | AB_2307391 | 1:20,000 | |
| Peroxidase AffiniPure rabbit anti-mouse IgG H&L | Jackson ImmunoResearch | 315-035-003 | AB_2340061 | 1:20,000 | |

WB, Western blotting; IF, immunofluorescence.

## Histone calculations

Absolute histone amounts were measured using mass spectrometry. The number of histones bound to a diploid zebrafish genome was calculated as 31,324,994 per genome for each histone. To arrive at this calculation (*Figure 4—source data 1*), we have previously shown that the average distance between the centers of neighboring nucleosomes in the zebrafish embryo around genome activation is 187 base pairs (*Zhang et al., 2014*). The size of a zebrafish genome is 1.46 Gb (GRCz10). This was multiplied by two to reach the diploid genome size which was then divided by the nucleosome repeat length to arrive at the number of nucleosomes per genome. As each histone is represented twice in a nucleosome, this number was multiplied by two to arrive at $3.13 \times 10^7$ copies of each histone that are required to wrap one zebrafish genome into chromatin (see *Figure 4—source data 1*). To arrive at numbers of histones in 'genomes worth of histones', the actual number of histones was divided by the amount of histones required to wrap one diploid genome. For the excess of histones per cell calculation (*Figure 4D*), the level of H2B (*Figure 4—source data 1*) and the cell numbers in *Figure 1—source data 1* were used. Because we have shown that all four core histones contribute to the repressor effect, we used the level of H2B for these calculations, as H2B is the lowest abundant histone and therefore may be limiting for the formation of nucleosomes. We subtracted the number of histones that are assumed to be bound to DNA, which amounts to one to two genomes worth of histones assuming replication (we used the average). The total concentration of non-DNA bound histones was calculated by dividing the total amount of non-DNA-bound histones by the volume of the animal cap (*Figure 4—figure supplement 1A*).

## Histone cocktail

Recombinant histones were of human origin and produced in *E. Coli* (NEB, Ipswich, MA; 1 mg/ml: H3.1 M2503S, H4 M2504S, H2A M2502S, H2B M2505S). We used human histones because histones are highly conserved between species and these are readily available. To remove DTT, H3.1 was dialyzed in histone buffer (300 mM NaCl, 1 mM EDTA, 20 mM NaPO$_4$, pH 7.0 at 25°C) using a Slide-A-Lyzer MINI dialysis device, 7K MWCO (ThermoFisher Scientific) at RT for 30 min or at 4°C overnight. Stoichiometric amounts of all four core histones were combined, spun for 5 min at 6600 rcf on a bench top centrifuge, supernatant was removed, and recovery of histones was measured via quantitative Western blot analysis and calculated using a standard. On average 5756 genomes worth of histone were injected with an error of ±388 (n = 3).

## NanoString analysis

This method involves assigning a unique color-coded barcode to transcripts of interest for single-molecule imaging and counting. The number of times the unique barcode is detected, is used as a readout of the expression level or number of 'counts' for the gene of interest. We developed a custom-made probe set of zygotically expressed genes as well as control genes (*Figure 2*, *Figure 2—source data 1*). Probe-sets were hybridized to 100 ng of mRNA extracted from a batch of 25 embryos using the RNeasy Mini Kit and processed following the manufacturer's recommendations

(NanoString Technologies, Seattle, WA) (*Kulkarni, 2011*). More information about the analysis can be found in the legend of *Figure 2—figure supplement 2*.

## Chromatin fractionation

At the desired stage, 65–100 embryos were manually deyolked and snap frozen in a cell lysis buffer (CLB: 10 mM HEPES pH 7.9, 10 mM KCl, 1.5 mM MgCl₂, 0.34 M sucrose, 10% glycerol, 0.1% NP-40, 1x protease inhibitor (Roche, Basel, Switzerland)) (*Méndez and Stillman, 2000*). Thawed embryos were shaken at 4°C for 5 min, then placed on ice and flicked intermittently for 5 min. Samples were spun in a bench top centrifuge at 1700 rcf for 5 min. Supernatant was removed and the pellet was washed with CLB. After another spin, the pellet was washed with a nuclear lysis buffer (3 mM EDTA, 0.2 mM EGTA). The sample was spun down and resuspended with high-salt solubilization buffer (50 mM Tris-HCL pH 8.0, 2.5 M NaCl, 0.05% NP-40, 1x protease inhibitor) (*Shechter et al., 2009*). The sample was vortexed for 2 min and placed on a rotator at RT for 10 min. The complete sample was then used in Western blot analysis.

## Co-immunoprecipitation

Per IP, 500 staged embryos were deyolked as previously described (*Link et al., 2006*). Cells were immediately resuspended in cell lysis buffer (10 mM Tris-HCl at pH 7.5, 10 mM NaCl, 0.5% NP-40, 1x protease inhibitor (Roche)), and lysed for 15 min on ice. Nuclei were pelleted by centrifugation and the supernatant was collected and rotated overnight at 4°C with 25 mL of protein G magnetic Dynabeads (Invitrogen, Carlsbad, CA) that had been pre-bound to an excess amount of antibody. Bound complexes were washed six times with RIPA buffer (50 mM HEPES at pH 7.6, 1 mM EDTA, 0.7% DOC, 1% Igepal, 0.5 M LiCl, 1x protease inhibitor) followed by 10 min of boiling in SDS loading buffer. Beads from the sample were subsequently removed by centrifugation and Western blotting was used for further analysis.

## Cy5 labeling

Histone H4 was incubated with Cyanine5 NHS ester (10:1 molar ratio) (Lumiprobe, Hannover, Germany) rotating overnight at 4°C. The next day, the solution was dialyzed in histone buffer for 30 min at RT. ~1 ng was injected into embryos of the Tg(h2afz:h2afz-GFP) transgenic fish line (*Pauls et al., 2001*) and embryos were imaged live on an upright LSM 510 META microscope equipped with a Zeiss W Plan-Apochromat 40 × 1.0 dipping objective. GFP was excited at 488 nm, detected with a PMT using BP527.5/545 and a pinhole size of 72 μm. Cy5 was excited at 633 nm, detected with the META detector using BP649-756 and a pinhole size of 96 μm. Images are 512*512 pixels, pixel size is 0.22 μm and were acquired with eight-bit mode.

## Quantification of nuclear concentration of non-DNA-bound histones

We determined the concentration of non-DNA bound histones in the nucleus as follows. First, we obtained volumetric data from live embryos, in which H4-sfGFP fusion protein was translated from injected mRNA to label animal cap and cell nuclei, at low- and high-intensity levels, respectively (see 'Live embryo tracking of nuclei and animal cap volumes' and 'Automated image analysis' below). Imaging live embryos prevented volume alterations due to fixation, permeabilization, and wash steps in immunofluorescence. Next, we determined relative histone distributions in cytoplasm and nucleus by immunofluorescence detection of endogenous histone H4, thus avoiding potential offsets or sub-cellular redistribution of the endogenous histone pool due to the addition of labeled fusion protein (see 'Immunofluorescence' and 'Automated image analysis' below). Then, we combined volumetric and nuclear-over-cytoplasmic intensity ratio data to allocate the total amount of histone H4 per embryo, as measured by mass spectrometry (*Figure 4—source data 1*), to the cytoplasmic and the nuclear sub-compartment (see 'Calculation of non-DNA-bound nuclear histone concentration' below). Lastly, aiming to calculate the concentration of only non-DNA-bound histones, the histones bound on chromatin in a given nucleus were subtracted from the total nuclear concentration of histones.

## Live embryo tracking of nuclei and animal cap volumes

To monitor the volumes of the animal cap and individual nuclei as well as nuclear import dynamics, histone H4 and PCNA were imaged in whole live embryos at a time resolution of 2 min or faster (see

Figure 4—source data 2). H4 was introduced as a fusion with sfGFP by mRNA injection. PCNA was monitored using offspring of transgenic fish with PCNA-RFP (Tg(bactin:RFP-pcna) [Strzyz et al., 2015]).

Embryos were mounted in glass capillaries with 1% low-melting agarose (Invitrogen) dissolved in 0.3x Danieau's solution and imaged with a Zeiss Z.1 lightsheet microscope using a 10x water dipping objective (NA 0.5) for acquisition, a lightsheet thickness below 5 µm, and dual side illumination (Icha et al., 2016). The microscopy chamber was filled with 0.3x Danieau's and kept at 28.5°C. Optical sectioning was 1 or 1.5 µm, time resolution was 2 min or faster for the acquisition of a whole 3D stack.

## Immunofluorescence

A time series of wild-type TLAB embryos covering 64-cell to sphere stages was collected, immunostained following a protocol optimized for full transparency and penetration of antibody, and imaged using a Zeiss Z.1 lightsheet microscope. Wild-type TLAB embryos were transferred at a given stage by transfer from 0.3x Danieau's into 2% formaldehyde in 0.3x Danieau's with 0.2% Tween-20 and left to fix overnight at 4°C. On the next day, embryos were washed three times for 10 min in PBST (Dulbecco's PBS with 0.1% Tween-20), then further permeabilized by washing twice in double-distilled water followed by 5 min waiting at room temperature, and then blocked with 4% BSA in PBST with 1% DMSO for at least 30 min. Primary antibodies against histone H4 and RNA polymerase II were diluted in 2% BSA in PBST with 1% DMSO and applied for incubation at 4°C for at least 48 hr. Embryos were washed three times for 10 min in PBST. Secondary antibodies were diluted in 2% BSA in PBST with 1% DMSO and applied overnight or longer at 4°C. Embryos were then washed three times for at least 10 min in PBST and stored at 4°C until imaging. Mounting for imaging was done in glass capillaries using 2% low-melting agarose dissolved in Dulbecco's PBS. 3D stacks were acquired using a 20x water dipping objective (NA 1.0) for acquisition, a lightsheet thickness below 5 µm, and dual side illumination. The microscopy chamber was filled with Dulbecco's PBS. Optical sectioning was 1 µm or less.

## Automated image analysis

Microscopy data were analyzed with a custom MatLab code using the Open Microscopy Environment bioformats plugin for stack reading. Nuclei were segmented using iterative thresholding for individual nuclei to compensate for differing intensities across the sample. 3D segments representing nuclei were dilated in two steps, giving a once- and a twice-extended shell around any given nucleus (Stasevich et al., 2014). The once-extended shell was removed from the twice-extended shell, along with any other nuclei that happened to be covered by the twice-extended shell. The resulting 3D segment thus covered cytoplasm in the vicinity of a given nucleus. The nucleus and the cytoplasm 3D segments were then used as masks to extract the mean intensity of a given cell's nucleus and cytoplasm. The animal cap was segmented in 3D based a single, global threshold determined from maximum intensity z-projections using Otsu's method. (For code, see Hilbert L. 2016 GitHub. https://github.com/lhilbert/NCRatio_Analysis. a7a5849).

For live-imaging data, individual nuclei were tracked across consecutive time frames based on minimal centroid distances. The volume fraction of the animal cap taken up by nuclei was calculated from the sum of volumes of all nuclei detected in a given stage, at their individual times of maximal extension in the respective cell cycle. (For code, see Hilbert L. 2016 GitHub. https://github.com/lhilbert/NucCyto_Ratio_TimeLapse. 55ed0fc).

For immunofluorescence data, nuclei were segmented based on the Pol II signal, which exhibited strong nuclear localization during interphase for all stages. Nuclear and cytoplasmic intensities for both Pol II and H4 were then extracted based on the Pol II segmentation as described above. To remove nuclei that were not in interphase or suffered signal degradation due to excessive spherical aberration or out-of-focus light, only nuclei with a nuclear-over-cytoplasmic intensity ration of greater than two were included in the analysis. Intensity ratios were strongly affected by background staining, so that H4 intensity values were corrected by subtraction of background levels before calculating ratios. Background levels were obtained from control embryos incubated with secondary, but not primary antibodies, which were imaged in the same session and with the same settings as the fully stained samples.

## Calculation of non-DNA-bound nuclear histone concentration

To obtain nuclear histone concentration values, one considers that the total number of histones must correspond to the contributions from all cells' cytoplasm and nuclei,

$H_{total} = [H_{nuclear}] \times V_{nucleus}^{sum} + [H_{cytoplasm}] \times V_{cytoplasm}^{sum}$, where $H_{total}$, $H_{nuclear}$, $H_{cytoplasm}$ are the total, the nuclear, and the cytoplasmic concentration of endogenous histone H4, respectively. $V_{nucleus}^{sum}$ and $V_{cytoplasm}^{sum}$ are the summed volumes of all cells' nuclei and cytoplasm, respectively. Dividing both sides by the total animal cap volume, $V_{total}$, one finds

$$\frac{H_{total}}{V_{total}} = [H_{nuclear}] \times v + [H_{cytoplasm}] \times (1-v),$$

where $v = V_{nucleus}^{sum}/V_{total}$ is the fraction of the total animal cap volume taken up by nuclei (also corresponds to the average fraction of cell volume occupied by the cell nucleus). Considering the N/C intensity ratio, $R$, to represent the concentration ratio, $R \approx [H_{nuclear}]/[H_{cytoplasm}]$, one can solve for $[H_{nuclear}]$,

$$[H_{nuclear}] = \frac{H_{total}}{V_{total}} \times \frac{R}{1 + v(R-1)}.$$

Realizing that this measured nuclear concentration results from non-DNA-bound histones as much as chromatin bound histones, one needs to subtract the concentration of chromatin bound histones to arrive at the non-DNA-bound histone H4 concentration,

$$[H_{free}] = [H_{nucleus}] - [H_{bound}] = [H_{nucleus}] - \frac{g}{V_{nucleus}^{single}}.$$

$g$ quantifies the number of complete, histone wrapped genomes (in units of genomes worth) being present in the volume of an individual nucleus, $V_{nucleus}^{single}$. Dropping the *single* superscript for ease of notation, the final expression is

$$[H_{free}] = \frac{H_{total}}{V_{total}} \times \frac{R}{1 + v(R-1)} - \frac{g}{V_{nucleus}}.$$

We measured all variables except $g$ on the right hand side using mass spectrometry ($[H_{total}]$), light-sheet imaging of whole live embryos injected with mRNA for H4-sfGFP ($V_{total}$, $v$, $V_{nucleus}$, see above), or immunofluorescence of endogenous histone H4 ($R$) (see *Figure 4—figure supplement 1B*). $g$ was assigned a value of 1.5 genomes worth, to fall between 1 (before replication of the genome) and 2 (complete replication of the genome), under the assumption of full occupation of the DNA by histones.

## Chromatin immunoprecipitation

Per IP, ~550 staged embryos were fixed at RT for 15 min in 1.85% formaldehyde. The fixative was quenched with 125 mM glycine and rotation at RT for 5 min. Embryos were then rinsed 3x in ice cold PBS (Accugene, Willowbrook, IL), resuspended in cell lysis buffer (same as co-IP) and lysed for 15 min on ice. Nuclei were pelleted by centrifugation, resuspended in nuclear lysis buffer (50 mM Tris-HCl at pH 7.5, 10 mM EDTA, 1% SDS, 1x protease inhibitor) and lysed for 10 min on ice. Two volumes of IP dilution buffer (16.7 mM Tris-HCl at pH 7.5, 167 mM NaCl, 1.2 mM EDTA, 0.01% SDS, 1x protease inhibitor) was added and the sample was sonicated to produce DNA fragments of between 200 and 300 bp (for Pou5f3-2xHA ChIP) or 400 and 500 bp (for tTA-VP16-2xHA ChIP) as determined using a bioanalyzer. 0.8% Triton X was added to the chromatin, which was then centrifuged to remove residual cellular debris. A sample was saved for input and the rest was divided over 25 mL of protein G magnetic Dynabeads that had been pre-bound to an excess amount of either HA antibody or IgG control antibody (*Table 3*). These were rotated overnight at 4°C. Bound complexes were washed six times with RIPA buffer followed by TBS. Elution buffer (50 mM NaHCO3, 1% SDS) was added to the beads, which were then vortexed and incubated for 15 min at RT on a rotator. Elutant was collected after centrifugation at 13,200 rcf, and beads were subjected to a repetition of the elution step. The same volume of elution buffer was added to the input sample, and 300 mM NaCl was added to all samples to reverse crosslink at 65°C overnight. Three volumes of 100%

ethanol was added and samples were incubated for 1 hr at −80°C. Samples were spun at 13,200 rcf at 4°C for 10 min followed by supernatant removal and air drying. 100 µL water was added and samples were shaken at RT for 5 hr. A PCR purification kit (Qiagen) was used before qPCR analysis.

### Sample-size determination

A minimum of three biological replicates were used for each experiment, with most experiments having four or more biological replicates (see figure legends for sample size).

## Acknowledgements

This work was supported by MPI-CBG core funding, a Human Frontier Science Program Career Development Award (CDA00060/2012) to NLV, a fellowship of the Dresden International Graduate School for Biomedicine and Bioengineering (DIGS-BB) granted by the German Research Foundation (DFG) to SRJ, and an ELBE Postdoctoral Fellowship granted by the Center for Systems Biology Dresden to LH. We thank members of the Vastenhouw laboratory, F Buchholz, D Drechsel and K Steinberg-Bains for help and advice, J Brugues, M Krause, YT Lin, IK Patten, P Tomancak, A Tóth, and J Rink for discussions and critical reading of the manuscript, the Norden lab for kindly providing the Tg(bactin:RFP-pcna) fish and the following MPI-CBG Services and Facilities for their support: the Biomedical Services (Fish unit), the Light Microscopy Facility, and Scientific Computing.

## Additional information

### Funding

| Funder | Grant reference number | Author |
| --- | --- | --- |
| Deutsche Forschungsgemeinschaft | | Shai R Joseph |
| Max-Planck-Gesellschaft | | Shai R Joseph<br>Máté Pálfy<br>Lennart Hilbert<br>Jens Karschau<br>Vasily Zaburdaev<br>Andrej Shevchenko<br>Nadine L Vastenhouw<br>Mukesh Kumar |
| Human Frontier Science Program | CDA00060/2012 | Nadine L Vastenhouw<br>Máté Pálfy |

The funders had no role in study design, data collection and interpretation, or the decision to submit the work for publication.

### Author contributions

SRJ, Conceptualization, Resources, Data curation, Formal analysis, Investigation, Visualization, Methodology, Writing—review and editing; MP, Conceptualization, Data curation, Investigation, Visualization, Writing—review and editing; LH, Conceptualization, Software, Investigation, Visualization, Methodology, Writing—review and editing; MK, Conceptualization, Investigation, Methodology, Writing—review and editing; JK, Conceptualization, Investigation, Writing—review and editing; VZ, Conceptualization, Supervision, Funding acquisition, Investigation, Writing—review and editing; AS, Conceptualization, Supervision, Funding acquisition, Investigation, Methodology, Writing—review and editing; NLV, Conceptualization, Resources, Supervision, Funding acquisition, Investigation, Methodology, Writing—original draft, Project administration

### Author ORCIDs

Shai R Joseph, http://orcid.org/0000-0002-7361-026X
Lennart Hilbert, http://orcid.org/0000-0003-4478-5607
Nadine L Vastenhouw, http://orcid.org/0000-0001-8782-9775

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
