## [Decision Letter]

[Editors’ note: a previous version of this study was rejected after peer review, but the authors submitted for reconsideration. The first decision letter after peer review is shown below.]

Thank you for choosing to send your work entitled "Histone level determines the timing of zygotic genome activation in zebrafish" for consideration at *eLife*. Your full submission has been evaluated by Janet Rossant (Senior Editor) and three peer reviewers, one of whom is a member of our Board of Reviewing Editors, and the decision was reached after discussions between the reviewers. Based on our discussions and the individual reviews below, we regret to inform you that your work will not be considered further for publication in *eLife*.

As you will see, the reviewers broadly agreed on several key weaknesses in this work:

1) The extensive overlap with previous work;

2) The absence of a clear demonstration that adding excess histones alters nucleosome levels on DNA (or that depletion of H2B reduces nucleosome occupancy);

3) Model that perhaps over-simplifies the scenario. Nothing in this manuscript or previous work pinpoints nucleosomes as the repressors. It remains entirely possible that it is a factor that binds nucleosomes – and not the octamer itself – that inhibits transcription.

However, all reviewers also saw the potential in this system and encourage the authors to delve deeper into this interesting question to shed new light on zygotic genome activation.

*Reviewer #1:*

Work described in this manuscript tests the idea that histone levels define the timing of zygotic genome activation, using zebrafish as a model system.

Background on the question: The idea that core histones repress zygotic transcription in the early embryo dates back to the early 90s (Prioleau et al., Cell 1994) and was explored in detail in *Xenopus oocytes* and embryos (e.g. Almouzni and Wolffe, EMBO, 1995). These studies involved injection of excess DNA, core histones, TBP and the activator VP16, leading to the conclusion that transcriptional quiescence before zygotic genome activation was due to a general repressive effect of histones and a defect in transcription activation. Work from the Wolffe lab also noted that they had not established that core histones were directly responsible for repression, but that nucleosome-binding proteins such as linker histones could functionally carry out repression. This possibility was recently supported by work in *Drosophila* (Perez-Montero et al. Dev. Cell, 2013) which showed that a specific variant of linker histone H1 affected the timing of zygotic genome activation (see also nice preview entitled "A histone timer for zygotic genome activation, by Tamkun and colleague, Dev Cell, 2013).

The current manuscript explores the role of histones in zebrafish genome activation. Quantitative mass spec nicely measures the histone levels across early development, confirming their gross excess over DNA. Injection of a cocktail of core histones is found to delay gastrulation and the onset of zygotic transcription (as measured by qPCR of 6 transcripts). All core histones were required for a maximal delay in genome activation, indicating that the formation of an entire nucleosome is essential for full repression. The reciprocal experiments were also performed, wherein the level of histone H2B is reduced to decrease the potential for nucleosome formation. These experiments show a modest premature activation of zygotic transcription. The model proposed is well in line with that from the Wolffe lab, involving both the titration of inhibitory nucleosomes and the involvement of an activator that can overcome such repression at selected sites.

Overall, the manuscript is well written and the hypothesis being tested is clear. The experiments seem to be carefully performed and the results are generally convincing. I like the quantitative mass spec and the focus on activation of endogenous genes (previous work largely looked at activation of genes located on exogenous DNA). One caveat is that the effects of histone H2B depletion on gene activation appears small, and I would encourage the authors to consider trying to reduce more than one core histone to achieve stronger effects.

My primary concern is that the conclusions do not go much beyond the model established in the mid-90s in *Xenopus*, reducing the impact of this work. Further, the authors should acknowledge that their work does not define core histones as the repressors, only that repression requires nucleosome formation (as in earlier work). I see this as a missed opportunity, given the very nice system developed. Why not target linker histones, specific transcription activators or repressors? That would allow them to go beyond what has been shown in *Xenopus* and *Drosophila* to define direct repressors and activators in their system.

*Reviewer #2:*

In the manuscript "Histone level determines the timing of zygotic genome activation in the zebrafish", the authors demonstrate that excessive quantities of histone proteins repress transcription during the early stages of zebrafish development. When injected with even more excess quantities of histone proteins, development and onset of transcription is delayed. These findings show are exciting and suggest a clever mechanism for regulating the timing of transition from maternal to zygotic transcription. If the authors could address the following issues, the manuscript would be of interest to the readers of *eLife*.

1) The authors give a compelling argument for the role of excess nucleosomes in preventing zygotic transition, however some direct evidence of the excess nucleosomes actually interacting with DNA is necessary to support their data. Showing maintenance of nucleosomes on the tested loci upon overexpression of histones and loss of nucleosome occupancy when the H2B morpholino is used would provide some evidence of the effect of the excess nucleosomes. As well, using ChIP to assay binding of transcriptional machinery to show differences with excess or loss of histone proteins.

2) While the authors show a role for nucleosomes in the timing of transition, they do not address the idea that a repressive chromatin remodeling complex or some other factor may still contribute to the timing of transition as well. Hdac1 is a likely candidate for this role, but there are many repressors that could contribute. It would be helpful for the authors to test Hdac1 or other repressors or discuss the possibility that other repressors may exist.

3) Authors estimate that the entire genome is fully bound once every 170bp with histones, with no mention of possible variations. While this is obviously an estimate, it would be helpful to at least note that this is an estimate based on the expectation of full occupancy of the genome with histones during the entire course of experiments.

4) The histone calculations are a bit confusing, particularly with the phrase "2x1011 copies of each per embryo". It would be helpful to state histone proteins rather than copies. Is this actually per embryo or per cell? If it is per embryo, then the calculations for excess amounts of histones added for later experiments are confusing, since the measurement is taken at the 8 cell stage, so 8 there would be genomes worth of histones. It seems like the 1.8x1011 histone proteins should be divided by 8x the number of histones in one genome rather than one single genome, giving an excess of 636 rather than 5049.

5) Calculations of the fold excess are slightly different using the numbers in the paper. Using the numbers from the paper, the fold excess at the 1000 cell stage calculates to 705 rather than 715. 1.8x1011 histone proteins divided by 35,425,427 comes out to 5081 rather than 5049.

*Reviewer #3:*

The question of how a developing embryo controls the transition from reliance on maternally loaded mRNA to activation of zygotic expression is a central question in developmental biology. The current manuscript examines this phenomenon in the context of early zebrafish development. The authors hypothesize that maternally loaded histones compete with the transcriptional machinery and that zygotic transcription ensues upon reaching a critical ratio of DNA to histones. They take the general approach of manipulation of the histone:DNA ratio by (1) microinjection of exogenous recombinant human histones or (2) depletion of endogenous H2B using morpholino technology, assessing (1) transcriptional states via RT-PCR of selected protein-coding genes as well as (2) developmental progression.

These experiments are highly reminiscent of papers published about 20 years ago by the Mechali (Prioleau et al., (1994) Cell 77: 439-449) and Wolffe (Almouzni and Wolffe, (1995) EMBO J 14: 1752-1765) labs in which the histone:DNA ratio was manipulated in early *Xenopus* development via microinjection of DNA. Regardless of the difference in technique, the current manuscript and the Mechali/Wolffe studies come to an identical conclusion – that the ratio of histones to DNA in early development is a critical determinant of the onset of zygotic transcription. Mechali and colleagues state "Our data suggest that the large excess of histones represses gene activation during early development…". Given this precedent in the literature, it is unclear to me why the current manuscript states repeatedly that the importance of the histone:DNA ratio in zygotic gene activation has never been established (i.e. in Abstract – "the titration of a transcriptional repressor by exponentially increasing amounts of DNA has been suggested as a possible mechanism…, but the repressor has never been identified.").

Mechanistically, I do not understand what exactly the current manuscript is measuring. The authors purchase recombinant human histones (NEB). These are supplied as purified, single polypeptides – not reconstituted into octamers or any other physiologically relevant complex. They then mix the histones together and inject the cocktail into embryos. It is not clear to me what the fate of the injected histones would be in this scenario. Recombinant histones at low salt concentration will be largely unstructured, certainly not present as octamers (and likely not as H3/H4 tetramers or H2A/H2B heterodimers). I encourage the authors to provide biochemical evidence that injected histones can associate in biochemical species similar to existing histone pools in the embryo – which are largely associated with histone chaperones. The conclusion by the authors that nucleosomes are the relevant species in their experiments competing with transcription factors for DNA (as opposed to non-specific interactions of unstructured basic proteins) seems poorly supported by the data.

[Editors’ note: what now follows is the decision letter after the authors submitted for further consideration.]

Thank you for resubmitting your work entitled "Competition between histone and transcription factor binding regulates the onset of transcription in zebrafish embryos" for further consideration at *eLife*. Your revised article has been favorably evaluated by Robb Krumlauf (Senior Editor), a Reviewing Editor, and two reviewers. The manuscript has been considerably improved, and would now be suitable for publication once several major concerns below are addressed.

Summary:

By utilizing efficient quantitative techniques, Joseph et al. present a very interesting study that supports a competitive interaction between histones and transcription factors to set the time for ZGA. They conclude that histones and critical transcription factors compete in binding to DNA, and eventually, the concentration of non-DNA bound core histone decides the timing of ZGA during zebrafish gastrulation. By injecting stoichiometric mixture of all four core histones they showed that increasing their levels delays the onset of zygotic transcription. On the other hand, decreasing levels of histones, by sequestering H4 with PTX3, shows an opposite effect. Using a quantitative mass spectrometry approach, the authors calculated the number of non-DNA bound histones per cell in genomes worth of histone, and showed that this parameter drops dramatically as embryos approach gastrulation. This provides opportunity for critical transcription factors (such as Pou5f3) to bind to DNA and initiate transcription. Following this argument, increasing and decreasing the level of Pou5f3 predates and delays the onset of ZGA, respectively. Finally, in the context of a TRE-GFP reporter construct, by injecting either a heterologous transcription factor (rTA-VP16) alone, or by co-injecting it with histone cocktails, a ChIP-PCR analysis provided further evidence for the competitive interaction between histones and transcription factors to bind to TRE elements in the GFP reporter. Altogether, this study provides a detailed mechanistic analysis to answer a few important issues related to a long-standing concept in the field. The quantitative techniques followed in this study highlights the usefulness of zebrafish to answer critical questions which could be otherwise difficult in other model systems. Such analyses would be beneficial for the zebrafish community as a whole. Taken together, this would be suitable for publication in "*eLife*" if the following concerns could be addressed or explained properly in an updated manuscript.

Major points that must be addressed in a revised manuscript:

1) In Figure 5, the authors only tested Pou5f3 as an example transcription factor that is important for ZGA. The authors posit that 'specific' effects needed for ZGA are the result of competition with transcription factors, and suggest that only specific TFs could compete with histones (e.g. only those TF needed for activation of the first ZGA genes). It would be nice to see this idea better fleshed out. Did the authors try to knockdown other important developmental factors (like Pou5f1, *Sox2*, Nanog) to see the effect on ZGA? What about non-developmental factors? According to their model, the competition between ZGA-related transcription factors and histones should not be specific to Pou5f3 only. Rather, knockdown of any other pluripotency factor should have the same ZGA-related phenotype. The specificity/ generality of this phenomenon should be tested by knockdown of at least one additional TF.

2) The authors should clarify the nature of their competition model, through additional experiments, or clarifications in the text. At present (and as detailed below) it is not clear how the authors envision that excess histones are functioning, and the text and data are not consistent in this regard. For this work to have the desired impact, we encourage the authors to work towards clearly articulating and substantiating their model.

A) The reviewers were concerned about the absence of proof for a competition model in the data presented, and alternative interpretations for the data presented are plausible: for example, histone 'excess' may affect replication timing, thereby affecting timing of transcriptional activation at ZGA. While the integrated transgene and VP16 experiments is strong, one would like to see it 'working' in an endogenous target: at the very least, the same ChIP experiment performed in excess of histone should be done for the Apoeb gene for Pouf3t and Sox19 transcription factors. Otherwise, it is difficult to reconcile it with the specificity in competition program discussed by the authors in the subsection “Competition for DNA binding between histones and transcription factors regulates the onset of transcription”.

B) Perhaps an additional possibility to address whether the competition model is indeed occurring, would be to reduce histone chaperone activity at the 'early' 1K stage. This should increase non-DNA bound histones, and therefore should have the same effect as the injection cocktail.

C) If the effect on transcriptional activation is assumed to be due to non-DNA bound histones (subsection “Onset of transcription coincides with a reduction in nuclear histone concentration” and Summary) – why is the addition of Histone cocktail effective considering that the authors conclude that 'this led to efficient chromatin incorporation of the histones injected as determined using Cy5 labelling' (subsection “Increasing the levels of all core histones delays onset of transcription and gastrulation”)?

D) Along the same lines, in the Discussion, the authors state that, because the 4 core histones are necessary to see the effects of the injected cocktail on transcriptional activation, this suggests a role for the nucleosome for the repressor function of histones. This statement is not consistent with the interpretation that it is the 'non-DNA bound histones' that is effecting ZGA timing (e.g. nucleosome is composed by the histone octamer plus the DNA). Given the biochemical properties of the 4 histones, it is also unlikely that the histones injected remain as octamers in the cells.

E) Similarly, related to the PTX3 experiments (Figure 3, subsection “Decreasing the level of histones causes premature transcription”), the authors conclude that 'total levels of H3 and H4 are not affected upon PTX3 expression', but three lines below, they conclude that effective levels of histone H3 and H4 are decreased. While 'effective' may indicate that they are not necessarily free to be incorporated into chromatin, this experiment is also at odds with the hypothesis that 'non-DNA-bound histones' regulate ZGA timing.

F) The authors conclude that histones are effectively incorporated by doing immunofluorescence with Cy5-labelled H4. However, nuclear localisation is not a proof for incorporation – did the authors try the same immunofluorescence with e.g. either pre-extraction with tritron or salt?

3) In Figure 6, the authors check the occupancy of rTA-VP16 on TRE elements under histone cocktail (HC) overexpression. This experiment shows a decrease in rTA-VP16 level at TRE upon HC injection, but they did not comment if this leads to a concomitant increase in histone binding at TRE. It would be good to include a comment related to this issue in the respective Results section.

---

## [Author Response]

[Editors’ note: the author responses to the first round of peer review follow.]

*As you will see, the reviewers broadly agreed on several key weaknesses in this work:*

*1) The extensive overlap with previous work;*

We agree with the reviewers that previous work using injected species of exogenous DNA had suggested that histones might play a role in transcription regulation in the embryo. We apologize for not doing enough justice to this work and now clarified this part of the manuscript.

Questions that remained after previous studies, however, include:

Whether histones regulate the timing of zygotic transcription (transcription of endogenous genes in the embryo).If so, which histones are involved (important for understanding the mechanism).How this all should work (the mechanism).

In our revised manuscript, we address these questions. We first set up in an in vivo assay to analyze the onset of zygotic transcription (Figure 1) and show that all core histones are required to determine the timing of transcription in the embryo (Figure 2, Figure 3). Then we directly analyze the mechanism. We first quantified the levels of histones in the embryo, nucleus and on chromatin. We found a reduction in nuclear histone concentration that coincides with genome activation but this is not accompanied by a loss of nucleosome density on DNA. This led us to propose a model in which a competition for DNA binding between transcription factors and non-DNA bound histones regulates the onset of transcription (Figure 4). To test this model, we changed the levels of endogenous transcription factors and analyzed its effect on the timing of transcription. In agreement with our model, the levels of transcription factors also affect the timing of transcription (Figure 5). We further test the competition model using a heterologous transgene (Figure 6). Using this strategy, we could confirm the importance of transcription factor levels for the timing of transcription and we were able to show that histone levels affect the binding of a transcription factor. Taken together, we show that the onset of transcription in the zebrafish embryo is regulated by a competition for DNA binding between histones and transcription factors. Our study provides, to our knowledge, the first example of a developmental transition in which competition for DNA binding between histones and transcription factors plays an important role in transcriptional regulation.

*2) The absence of a clear demonstration that adding excess histones alters nucleosome levels on DNA (or that depletion of H2B reduces nucleosome occupancy);*

We show that histone levels regulate the time of zygotic transcription. Transcription starts when the nuclear concentration of histones drops, but this is not accompanied by a loss of nucleosome density on DNA. This suggests that the onset of transcription is regulated by the concentration of non-DNA bound histones in the nucleus. Based on this data, we propose that the onset of transcription in the embryo is regulated by a competition for DNA binding between non-DNA bound histones and transcription factors and we provide direct experimental evidence for this model. We therefore propose that the effect of changing histone levels experimentally is exerted through changes in the non-DNA bound concentration of histones and not by changes in global nucleosome density.

*3) Model that perhaps over-simplifies the scenario. Nothing in this manuscript or previous work pinpoints nucleosomes as the repressors. It remains entirely possible that it is a factor that binds nucleosomes – and not the octamer itself – that inhibits transcription.*

This is where we have really expanded the manuscript. Our experiments show that the four core histones mediate the repressor effect. Moreover, our finding that histones are neither completely depleted from the soluble fraction, nor generally depleted from chromatin, argues against a model in which a global loss of nucleosome density on chromatin causes the onset of transcription. This strongly suggests that histones themselves function as the actual repressors of transcription and argues against a role for the depletion of proteins associated with core histones. Furthermore, the model presented in the earlier submission did indeed simplify the scenario. We now present data that the levels of activators are also important. We provide direct evidence for a model in which competition for DNA binding between histones and activators regulates the onset of transcription in the embryo.

*However, all reviewers also saw the potential in this system and encourage the authors to delve deeper into this interesting question to shed new light on zygotic genome activation.*

*Reviewer #1:*

*Work described in this manuscript tests the idea that histone levels define the timing of zygotic genome activation, using zebrafish as a model system.*

*Background on the question: The idea that core histones repress zygotic transcription in the early embryo dates back to the early 90s (Prioleau et al., Cell 1994) and was explored in detail in Xenopus oocytes and embryos (e.g. Almouzni and Wolffe, EMBO, 1995). These studies involved injection of excess DNA, core histones, TBP and the activator VP16, leading to the conclusion that transcriptional quiescence before zygotic genome activation was due to a general repressive effect of histones and a defect in transcription activation. Work from the Wolffe lab also noted that they had not established that core histones were directly responsible for repression, but that nucleosome-binding proteins such as linker histones could functionally carry out repression. This possibility was recently supported by work in Drosophila (Perez-Montero et al. Dev. Cell, 2013) which showed that a specific variant of linker histone H1 affected the timing of zygotic genome activation (see also nice preview entitled "A histone timer for zygotic genome activation, by Tamkun and colleague, Dev Cell, 2013).*

We agree with the reviewer that in our previous manuscript, we did not do enough justice to the work that has previously shown a role for histones in repressing transcription in the early *Xenopus* embryo. We apologize and now discuss existing work more clearly and identify the aspects which needed further work.

*The current manuscript explores the role of histones in zebrafish genome activation. Quantitative mass spec nicely measures the histone levels across early development, confirming their gross excess over DNA. Injection of a cocktail of core histones is found to delay gastrulation and the onset of zygotic transcription (as measured by qPCR of 6 transcripts). All core histones were required for a maximal delay in genome activation, indicating that the formation of an entire nucleosome is essential for full repression. The reciprocal experiments were also performed, wherein the level of histone H2B is reduced to decrease the potential for nucleosome formation. These experiments show a modest premature activation of zygotic transcription. The model proposed is well in line with that from the Wolffe lab, involving both the titration of inhibitory nucleosomes and the involvement of an activator that can overcome such repression at selected sites.*

*Overall, the manuscript is well written and the hypothesis being tested is clear. The experiments seem to be carefully performed and the results are generally convincing. I like the quantitative mass spec and the focus on activation of endogenous genes (previous work largely looked at activation of genes located on exogenous DNA). One caveat is that the effects of histone H2B depletion on gene activation appears small, and I would encourage the authors to consider trying to reduce more than one core histone to achieve stronger effects.*

As suggested, we have changed our approach to reduce histones. Since we see by mass spectrometry that a large amount of histones is present as protein from fertilization on (Figure 4), we decided to change our approach and target protein levels. Now we use PTX3 to target histone proteins. We show in vivo that PTX3 interacts with H4 protein, and the effect of PTX3 is greater than with the morpholino.

We also expand our Discussion, and indicate that due to a role of activator levels in our competition model, pushing genome activation back will always be harder than forward (with more histones). Activators might be limiting because they need to be translated (Pou5f3, Nanog, *Sox2* are highly translated during development) and because many of these activators might function cooperatively, they all need to be present at the right concentration at the right time.

*My primary concern is that the conclusions do not go much beyond the model established in the mid-90s in Xenopus, reducing the impact of this work. Further, the authors should acknowledge that their work does not define core histones as the repressors, only that repression requires nucleosome formation (as in earlier work). I see this as a missed opportunity, given the very nice system developed. Why not target linker histones, specific transcription activators or repressors? That would allow them to go beyond what has been shown in Xenopus and Drosophila to define direct repressors and activators in their system.*

With respect to the novelty of our work, and the identity of the repressor, we refer to the key points.

*Reviewer #2:*

*In the manuscript "Histone level determines the timing of zygotic genome activation in the zebrafish", the authors demonstrate that excessive quantities of histone proteins repress transcription during the early stages of zebrafish development. When injected with even more excess quantities of histone proteins, development and onset of transcription is delayed. These findings show are exciting and suggest a clever mechanism for regulating the timing of transition from maternal to zygotic transcription. If the authors could address the following issues, the manuscript would be of interest to the readers of eLife.*

*1) The authors give a compelling argument for the role of excess nucleosomes in preventing zygotic transition, however some direct evidence of the excess nucleosomes actually interacting with DNA is necessary to support their data. Showing maintenance of nucleosomes on the tested loci upon overexpression of histones and loss of nucleosome occupancy when the H2B morpholino is used would provide some evidence of the effect of the excess nucleosomes. As well, using ChIP to assay binding of transcriptional machinery to show differences with excess or loss of histone proteins.*

To address these points, we did the following:

We have chemically tagged H4 and show that it incorporates into chromatin (Figure 2—figure supplement 1). We have done chromatin fractionation and show that nucleosome density does not change globally at the time of genome activation (Figure 4). In our competition model, local changes in nucleosome density (where RNA pol II or TF bind) result in transcription.

Exactly as suggested, we used ChIP-qPCR to assess the binding of a transcriptional activator in response to changing histone levels. Increasing histone levels causes a reduction of binding of the activator to its target site (Figure 6). In agreement with this, increasing activator levels leads to premature transcription (Figure 5, Figure 6).

*2) While the authors show a role for nucleosomes in the timing of transition, they do not address the idea that a repressive chromatin remodeling complex or some other factor may still contribute to the timing of transition as well. Hdac1 is a likely candidate for this role, but there are many repressors that could contribute. It would be helpful for the authors to test Hdac1 or other repressors or discuss the possibility that other repressors may exist.*

Here, we show that histone levels in combination with transcription factor levels are sufficient to change the timing of transcription. We agree with the reviewer, however, that chromatin remodeling complexes might also contribute to genome activation. We have now included a discussion on how the activity of chromatin remodelers might also be mediated by histone levels.

*3) Authors estimate that the entire genome is fully bound once every 170bp with histones, with no mention of possible variations. While this is obviously an estimate, it would be helpful to at least note that this is an estimate based on the expectation of full occupancy of the genome with histones during the entire course of experiments.*

The number we used is based on MNase-Seq experiments and the average nucleosome repeat length derived from that (Zhang et al., 2014). We have clarified this in the text. Of note is that the nucleosome repeat length that we use is not critical for our model. Even variations of nucleosome occupancy from 160 to 200 do not substantially affect our calculations of the total nor nuclear histone concentration.

*4) The histone calculations are a bit confusing, particularly with the phrase "2x1011 copies of each per embryo". It would be helpful to state histone proteins rather than copies. Is this actually per embryo or per cell? If it is per embryo, then the calculations for excess amounts of histones added for later experiments are confusing, since the measurement is taken at the 8 cell stage, so 8 there would be genomes worth of histones. It seems like the 1.8x1011 histone proteins should be divided by 8x the number of histones in one genome rather than one single genome, giving an excess of 636 rather than 5049.*

We apologize for the confusion and provide more clarity with our calculations in the new manuscript (Materials and methods and Table 2).

*5) Calculations of the fold excess are slightly different using the numbers in the paper. Using the numbers from the paper, the fold excess at the 1000 cell stage calculates to 705 rather than 715. 1.8x1011 histone proteins divided by 35,425,427 comes out to 5081 rather than 5049.*

This discrepancy was due to rounding our numbers. There is more clarity in our new Table 2.

*Reviewer #3:*

*The question of how a developing embryo controls the transition from reliance on maternally loaded mRNA to activation of zygotic expression is a central question in developmental biology. The current manuscript examines this phenomenon in the context of early zebrafish development. The authors hypothesize that maternally loaded histones compete with the transcriptional machinery and that zygotic transcription ensues upon reaching a critical ratio of DNA to histones. They take the general approach of manipulation of the histone:DNA ratio by (1) microinjection of exogenous recombinant human histones or (2) depletion of endogenous H2B using morpholino technology, assessing (1) transcriptional states via RT-PCR of selected protein-coding genes as well as (2) developmental progression.*

For a summary of the data in the revised manuscript we refer to the key points.

*These experiments are highly reminiscent of papers published about 20 years ago by the Mechali (Prioleau et al., (1994) Cell 77: 439-449) and Wolffe (Almouzni and Wolffe, (1995) EMBO J 14: 1752-1765) labs in which the histone:DNA ratio was manipulated in early Xenopus development via microinjection of DNA. Regardless of the difference in technique, the current manuscript and the Mechali/Wolffe studies come to an identical conclusion – that the ratio of histones to DNA in early development is a critical determinant of the onset of zygotic transcription. Mechali and colleagues state "Our data suggest that the large excess of histones represses gene activation during early development…". Given this precedent in the literature, it is unclear to me why the current manuscript states repeatedly that the importance of the histone:DNA ratio in zygotic gene activation has never been established (i.e. in Abstract – "the titration of a transcriptional repressor by exponentially increasing amounts of DNA has been suggested as a possible mechanism…, but the repressor has never been identified.").*

We apologize for not clearly reviewing existing data previously, and included a thorough discussion of existing work in our revised manuscript.

With respect to the novelty of our work, we refer to the key points.

*Mechanistically, I do not understand what exactly the current manuscript is measuring. The authors purchase recombinant human histones (NEB). These are supplied as purified, single polypeptides – not reconstituted into octamers or any other physiologically relevant complex. They then mix the histones together and inject the cocktail into embryos. It is not clear to me what the fate of the injected histones would be in this scenario. Recombinant histones at low salt concentration will be largely unstructured, certainly not present as octamers (and likely not as H3/H4 tetramers or H2A/H2B heterodimers). I encourage the authors to provide biochemical evidence that injected histones can associate in biochemical species similar to existing histone pools in the embryo – which are largely associated with histone chaperones. The conclusion by the authors that nucleosomes are the relevant species in their experiments competing with transcription factors for DNA (as opposed to non-specific interactions of unstructured basic proteins) seems poorly supported by the data.*

To address these points, we did the following:

We provide proof that the human NEB histones are functional. We have chemically tagged H4 that shows that it localizes to the nucleus and incorporates into chromatin (Figure 2—figure supplement 1).

Additionally, the experiments that remove one histone from the cocktail and elevate the levels of the other three, argue against a non-specific effect of an unstructured basic protein. If the delay in transcription was a result of randomly injecting unstructured protein, then the histone minus one experiment should also have the same effect. Which they don’t.

[Editors' note: the author responses to the re-review follow.]

*Major points that must be addressed in a revised manuscript:*

*1) In Figure 5, the authors only tested Pou5f3 as an example transcription factor that is important for ZGA. The authors posit that 'specific' effects needed for ZGA are the result of competition with transcription factors, and suggest that only specific TFs could compete with histones (e.g. only those TF needed for activation of the first ZGA genes). It would be nice to see this idea better fleshed out. Did the authors try to knockdown other important developmental factors (like Pou5f1, Sox2, Nanog) to see the effect on ZGA? What about non-developmental factors? According to their model, the competition between ZGA-related transcription factors and histones should not be specific to Pou5f3 only. Rather, knockdown of any other pluripotency factor should have the same ZGA-related phenotype. The specificity/ generality of this phenomenon should be tested by knockdown of at least one additional TF.*

We apologize for the confusion. We agree with the reviewers that, according to our model, the competition between ZGA-related transcription factors and histones should not be specific to Pou5f3. We have clarified this in the text.

Following the reviewers’ suggestion, we now show that knockdown of Sox19b and FoxH1 also results in a delay in the transcription of target genes (subsection “Decreasing transcription factor levels delays the onset of transcription”, last paragraph, Figure 5—figure supplement 1).

To further substantiate the generality of the role of histone levels in genome activation, we have also included NanoString analysis of embryos injected with the histone cocktail (subsection “Increasing the levels of all core histones delays onset transcription and gastrulation”, second paragraph, Figure 2 and Figure 2—figure supplement 2). This shows that transcription of an important number of genes is delayed as a consequence of increased histone levels. Given that these genes are likely regulated by a variety of transcription factors, this supports the notion that histone levels affect the binding of many transcription factors.

*2) The authors should clarify the nature of their competition model, through additional experiments, or clarifications in the text. At present (and as detailed below) it is not clear how the authors envision that excess histones are functioning, and the text and data are not consistent in this regard. For this work to have the desired impact, we encourage the authors to work towards clearly articulating and substantiating their model.*

Thank you for raising this point. We have detailed below what experiments we have done and what changes we have made to the text to address it.

*A) The reviewers were concerned about the absence of proof for a competition model in the data presented, and alternative interpretations for the data presented are plausible: for example, histone 'excess' may affect replication timing, thereby affecting timing of transcriptional activation at ZGA.*

That is a good point. Replication timing is not affected in our experiments, and we have now clarified this in the text (For example, subsection “Increasing the levels of all core histones delays onset transcription and gastrulation”, second paragraph and subsection “Decreasing the pool of available histones causes premature transcription”, just to mention a few).

*While the integrated transgene and VP16 experiments is strong, one would like to see it 'working' in an endogenous target: at the very least, the same ChIP experiment performed in excess of histone should be done for the Apoeb gene for Pouf3t and Sox19 transcription factors. Otherwise, it is difficult to reconcile it with the specificity in competition program discussed by the authors in the subsection “Competition for DNA binding between histones and transcription factors regulates the onset of transcription”.*

This is an excellent point. We used the transgene because the regulation of endogenous genes is complex and the data is therefore harder to interpret. However, we now show that the binding of Pou5f3 on apoeb (and dusp6, an additional gene for which we saw premature transcription upon Pou5f3/Sox19b overexpression), is also reduced in the presence of more histones (subsection “Transcription factor binding is sensitive to histone levels”, Figure 5).

*B) Perhaps an additional possibility to address whether the competition model is indeed occurring, would be to reduce histone chaperone activity at the 'early' 1K stage. This should increase non-DNA bound histones, and therefore should have the same effect as the injection cocktail.*

This is in principle a good idea. However, the lack of information on chaperones in zebrafish, as well as the – related – lack of specific drugs make this experiment difficult to perform and interpret. Especially in combination with the fact that in the best-case scenario this experiment would recapitulate the histone cocktail effect without providing more (quantitative) insight in the competition model, we decided not to try these experiments.

*C) If the effect on transcriptional activation is assumed to be due to non-DNA bound histones (subsection “Onset of transcription coincides with a reduction in nuclear histone concentration” and Summary) – why is the addition of Histone cocktail effective considering that the authors conclude that 'this led to efficient chromatin incorporation of the histones injected as determined using Cy5 labelling' (subsection “Increasing the levels of all core histones delays onset of transcription and gastrulation”)?*

This experiment was designed to show that the histones we inject can incorporate in chromatin, and thus become part of, and behave like, the endogenous pool of histones available for competition. Like endogenous histones in the early embryo, not all histones will incorporate into chromatin due to their excess levels. We have now clarified this in text (subsection “Increasing the levels of all core histones delays onset transcription and gastrulation”, second paragraph).

*D) Along the same lines, in the Discussion, the authors state that, because the 4 core histones are necessary to see the effects of the injected cocktail on transcriptional activation, this suggests a role for the nucleosome for the repressor function of histones. This statement is not consistent with the interpretation that it is the 'non-DNA bound histones' that is effecting ZGA timing (e.g. nucleosome is composed by the histone octamer plus the DNA). Given the biochemical properties of the 4 histones, it is also unlikely that the histones injected remain as octamers in the cells.*

We apologize for the confusion. Our model is based on the knowledge that the four core histones are required to form a histone octamer (and nucleosome). Of course, the histones only form a histone octamer when associating with DNA and it is therefore expected that the concentration of soluble histones dictates how likely a nucleosome is formed on an available piece of DNA. Thus, we propose that the soluble histones compete with transcription factors for binding, but ultimately it is either the histone octamer or transcription factor that occupies DNA. We have clarified this in the text (throughout).

*E) Similarly, related to the PTX3 experiments (Figure 3, subsection “Decreasing the level of histones causes premature transcription”), the authors conclude that 'total levels of H3 and H4 are not affected upon PTX3 expression', but three lines below, they conclude that effective levels of histone H3 and H4 are decreased. While 'effective' may indicate that they are not necessarily free to be incorporated into chromatin, this experiment is also at odds with the hypothesis that 'non-DNA-bound histones' regulate ZGA timing.*

In order to compete for DNA binding, histones need to be available (see also above). Therefore, when histones are not available to bind to DNA (because they are bound by PTX3), this lowers their effective concentration. We have clarified this in the text (subsection “Decreasing the pool of available histones causes premature transcription”).

*F) The authors conclude that histones are effectively incorporated by doing immunofluorescence with Cy5-labelled H4. However, nuclear localisation is not a proof for incorporation – did the authors try the same immunofluorescence with e.g. either pre-extraction with tritron or salt?*

We apologize for the confusion. The conclusion that histones are incorporated was not based on the nuclear localization of the labeled histone but rather on the fact that we see the label in chromatin during metaphase. We have now clarified this in the text (subsection “Increasing the levels of all core histones delays onset transcription and gastrulation”, second paragraph) and in Figure 2—figure supplement 1.

*3) In Figure 6, the authors check the occupancy of rTA-VP16 on TRE elements under histone cocktail (HC) overexpression. This experiment shows a decrease in rTA-VP16 level at TRE upon HC injection, but they did not comment if this leads to a concomitant increase in histone binding at TRE. It would be good to include a comment related to this issue in the respective Results section.*

We assume that there is an increase in histone binding but we have not looked at this. We have now clarified this in the text (subsection “Transcription factor binding is sensitive to histone levels”) in the section on Pou5f3 binding (Figure 5), which is the first time we test the effect of histone levels on transcription factor binding in the revised version of the manuscript.